# Zika virus infection induces host inflammatory responses by facilitating NLRP3 inflammasome assembly and interleukin-1β secretion

Wenbiao Wang[1,2], Geng Li[3], De Wu[4], Zhen Luo[2], Pan Pan[1], Mingfu Tian[1], Yingchong Wang[1], Feng Xiao[1], Aixin Li[1], Kailang Wu[1], Xiaohong Liu[3], Lang Rao[5], Fang Liu[1], Yingle Liu[1,2] & Jianguo Wu [1,2]

Zika virus (ZIKV) infection is a public health emergency and host innate immunity is essential for the control of virus infection. The NLRP3 inflammasome plays a key role in host innate immune responses by activating caspase-1 to facilitate interleukin-1β (IL-1β) secretion. Here we report that ZIKV stimulates IL-1β secretion in infected patients, human PBMCs and macrophages, mice, and mice BMDCs. The knockdown of NLRP3 in cells and knockout of NLRP3 in mice inhibit ZIKV-mediated IL-1β secretion, indicating an essential role for NLRP3 in ZIKV-induced IL-1β activation. Moreover, ZIKV NS5 protein is required for NLRP3 activation and IL-1β secretion by binding with NLRP3 to facilitate the inflammasome complex assembly. Finally, ZIKV infection in mice activates IL-1β secretion, leading to inflammatory responses in the mice brain, spleen, liver, and kidney. Thus we reveal a mechanism by which ZIKV induces inflammatory responses by facilitating NLRP3 inflammasome complex assembly and IL-1β activation.

[1] State Key Laboratory of Virology and College of Life Sciences, Wuhan University, Wuhan, 430072, P.R. China. [2] Institute of Medical Microbiology, Jinan University, Guangzhou, 510632, P.R. China. [3] School of Chinese Pharmaceutical Science, Guangzhou University of Chinese Medicine, Guangzhou, 510006, P.R. China. [4] Institute of Pathogenic Microbiology, Center for Disease Control and Prevention of Guangdong, Guangzhou, 510006, P.R. China. [5] School of Physics and Technology, Wuhan University, Wuhan, 430072, P.R. China. Wang W, Li G, Wu D and Luo Z contributed equally to this work. Correspondence and requests for materials should be addressed to F.L. (email: liufung@whu.edu.cn) or to Y.L. (email: mvlwu@whu.edu.cn) or to J.W. (email: jwu@whu.edu.cn)

Outbreak of Zika virus (ZIKV) in South America constitutes a public health emergency[1,2]. The virus was originally discovered in 1947 in a febrile Rhesus macaque caged in Uganda[3]. It typically causes a mild and self-limiting illness known as Zika fever, which is accompanied by maculopapular rash, headache, and myalgia. The 2007 outbreak in the Federated States of Micronesia affected about 75% of the population[4]. ZIKV infection expanded into North America in 2014 and 2015[5,6]. It is now an international concern due to its link to devastating neurological complications, including Guillain–Barre Syndrome and meningoencephalitis in adults and microcephaly in fetuses[7–9], along with testis damage and male infertility in mice[10,11]. ZIKV genome encodes a single polypeptide that is proteolytically processed by viral and host proteases to produce three structural proteins (capsid C, premembrane prM, and envelope E) and seven non-structural proteins (NS1, NS2A, NS2B, NS3, NS4A, NS4B, and NS5)[12]. The NS5 protein is a viral RNA-dependent RNA polymerase (RdRp), which synthesizes viral genomic RNA via a de novo initiation mechanism[13].

The host innate immune system detects viral infection by recognizing molecular patterns[14]. The best-characterized viral sensors are pattern-recognition receptors, including Toll-like receptors[15], RIG-I-like receptors[16], NOD-like receptors (NLRs)[17] and C-type lectin receptors[18]. The NLRs are involved in the assembly of large protein complexes known as inflammasomes, which are involved in the innate immune response to pathogens[19]. Inflammasomes consist of a cytoplasmic sensor molecule (such as NACHT, LRR, and PYD domain-containing protein 3 (NLRP3)), the adaptor protein (apoptosis-associated speck-like protein containing caspase recruitment domain (ASC)), and the effecter protein (pro-Caspase-1). NLRP3 and ASC promote pro-Casp-1 cleavage to generate the active subunits p22 and p20, leading to the maturation and secretion of interleukin-1β (IL-1β)[20]. IL-1β plays crucial roles in inflammatory responses, instructs adaptive immune responses by inducing expression of immunity-associated genes, and facilitates lymphocyte recruitment to the site of infection[21,22].

Here we reveal a mechanism by which ZIKV infection induces host inflammatory responses by facilitating the NLRP3 inflammasome assembly and IL-1β secretion. Clinical investigations, animal analyses, and cellular studies show that ZIKV induces IL-1β secretion by activating the NLRP3 inflammasome. Interestingly, ZIKV NS5 directly binds NLRP3 to facilitate the assembly of NLRP3 inflammasome complex by forming a sphere-like structure of NS5–NLRP3–ASC. Moreover, ZIKV induces considerable inflammatory responses in the brain, spleen, liver, and kidney of infected mice. Thus we report a function of ZIKV NS5 in regulating the NLRP3 inflammasome and reveal a mechanism by which ZIKV induces host inflammatory and immune responses.

## Results

**ZIKV infection activates IL-1β production and secretion.** ZIKV first appeared in South China in 2016 with at least 22 infected cases reported, most of them imported[23,24]. Here we initially showed that IL-1β levels in the sera of ZIKV-infected patients ($n = 11$) were higher than those in healthy individuals ($n = 13$) (Fig. 1a), suggesting that ZIKV infection is associated with IL-1β secretion. The correlation between ZIKV infection and IL-1β secretion was evaluated in A129 mice deficient in type I receptors[25]. In the blood of infected mice, the viral titers peaked at 2 days postinfection and then declined (Fig. 1b), and in the sera of infected mice, the IL-1β levels increased rapidly until 4 days postinfection and decreased thereafter (Fig. 1c), demonstrating that ZIKV induces IL-1β production and secretion.

The effect of ZIKV on IL-1β activation was then determined. In human peripheral blood mononuclear cells (PBMCs), *IL-1β* mRNA expression (Fig. 1d, f) and protein secretion (Fig. 1e, g) were stimulated by lipopolysaccharides (LPS) and ZIKV. In phorbol-12-myristate-13-acetate (TPA)-differentiated THP-1 macrophages[26], *IL-1β* mRNA was activated by ZIKV but not by Nigericin (an NLRP3 activator) (Fig. 1h, j), IL-1β secretion was induced by Nigericin and ZIKV (Fig. 1I, k), IL-1β maturation and Casp-1 cleavage in cell supernatants, and pro-IL-1β production in cell lysates were activated by Nigericin and ZIKV (Fig. 1l, m). Moreover, in bone marrow dendritic cells (BMDCs) differentiated from C57BL/6 mice, *IL-1β* mRNA and protein levels were induced by LPS+Nigericin and ZIKV (Fig. 1n, o). ZIKV RNA was expressed in infected PBMCs and THP-1 cells (Supplementary Fig. 1a–d), infectious ZIKV was detected in the cell supernatant of PBMCs and THP-1 cells (Supplementary Fig. 1e, f), ZIKV E protein was produced in THP-1 cells (Supplementary Fig. 1g), and ZIKV RNA was detected in mice BMDCs (Supplementary Fig. 1h), indicating that ZIKV is replicated well in the infected cells. Taken together, we demonstrate that ZIKV activates the production and secretion of IL-1β in infected patients and cultured cells.

**ZIKV activates NLRP3 inflammasome to induce IL-1β secretion.** The activation of IL-1β is regulated by two pathways: the transcription of *pro-IL-1β* mRNA regulated by nuclear factor (NF)-κB and the procession of IL-1β mediated by Casp-1[19]. In THP-1-differentiated macrophages, IL-1β secretion was stimulated by Nigericin and ZIKV, but this activation was repressed by VX-765 (Casp-1 inhibitor) (Fig. 2a). Similarly, IL-1β and Casp-1 cleavages were activated by Nigericin and ZIKV, but such activations were repressed by VX-765. However, the levels of pro-IL-1β and pro-Casp-1 proteins were not affected by VX-765 (Fig. 2b). These results suggest that Casp-1 is involved in ZIKV-induced activation of IL-1β.

To assess the role of the NLRP3 inflammasome in ZIKV-induced secretion of IL-1β, a THP-1 cell line that stably expressed short hairpin RNAs (shRNAs) specific to the genes (*NLRP3*, *ASC*, and *Casp-1*) encoding for the NLRP3 components was generated based on previous reports[27–29]. In these cells, IL-1β secretion, IL-1β maturation, and Casp-1 cleavage were induced by Nigericin and ZIKV, but these activations were attenuated by *sh-NLRP3*, *sh-ASC*, and *sh-Casp-1* (Fig. 2c–f). Thus knockdown of NLRP3 inflammasome components attenuates ZIKV-induced activation of IL-1β.

The role of NLRP3 in ZIKV-induced secretion of IL-1β was further determined in C57BL/6 WT mice and *NLRP3*$^{-/-}$ mice. The level of IL-1β was significantly stimulated by LPS+adenosine triphosphate (ATP) and ZIKV in BMDCs of C57BL/6 mice but not in BMDCs of *NLRP3*$^{-/-}$ mice (Fig. 2g). The production of NLRP3 protein was confirmed in the BMDCs of C57BL/6 mice, but not in *NLRP3*$^{-/-}$ mice (Fig. 2h). ZIKV RNA was detected in the BMDCs of infected mice, but not in LPS+ATP-treated mice (Supplementary Fig. 2a). Similarly, IL-1β level was significantly induced by LPS+ATP and ZIKV in BMDMs of C57BL/6 mice, but not in *NLRP3*$^{-/-}$ mice (Fig. 2i). ZIKV RNA was detected in the BMDMs of infected mice, but not in LPS+ATP-treated mice (Supplementary Fig. 2b). Thus we reveal that NLRP3 is required for ZIKV-induced activation of IL-1β and that ZIKV induces IL-1β secretion by activating the NLRP3 inflammasome.

The localization of NLRP3 as specks is an indication of inflammasome complex formation[20]. We thus investigated the effect of ZIKV on NLRP3 inflammasome formation. NLRP3 was diffusely distributed in the cytoplasm of uninfected cells but formed distinct small specks in ZIKV-infected THP-1-

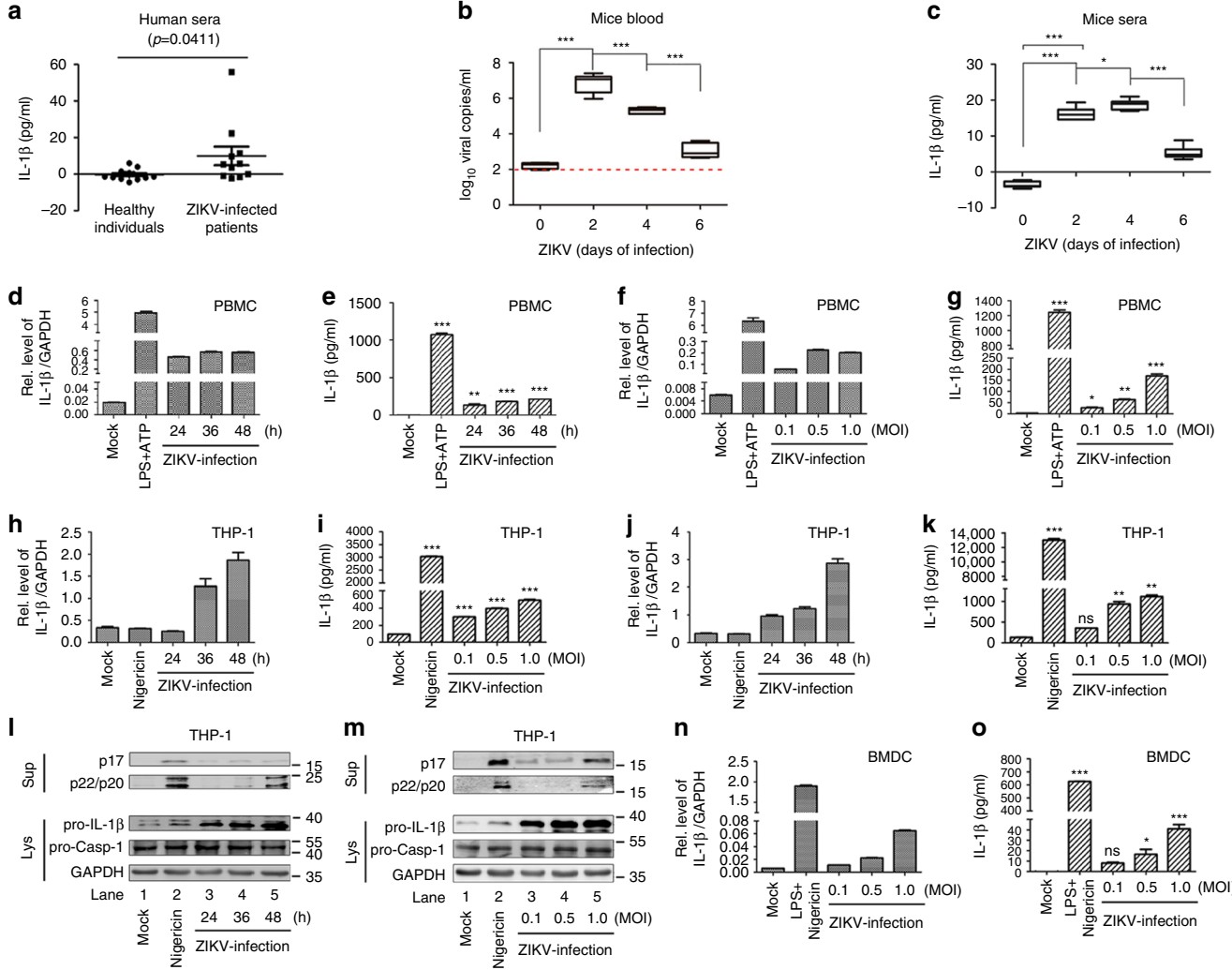

**Fig. 1** The effects of ZIKV infection on IL-1β production and secretion. **a** IL-1β levels in the sera of patients ($n = 11$) and healthy individuals ($n = 13$) was determined by ELISA. Data shown are means ± s.e.m; *$P < 0.05$ (two-tailed Student's $t$-test). **b, c** Six-week-old A129 mice ($n = 7$; 4 males and 3 females) were infected with ZIKV ($5 \times 10^5$ PFU) for 0, 2, 4, and 6 days. Viral titers in the blood were determined by RT-PCR (**b**). IL-1β levels in the sera were determined by ELISA (**c**). Data shown are whiskers: min.–max.; *$P < 0.05$, ***$P < 0.0001$ (one-way ANOVA with Tukey's post-hoc test). **d–g** PBMCs isolated from healthy individuals were treated with LPS (1 μg/ml) for 6 h or 2 μM Nigericin for 2 h or infected with ZIKV at an MOI = 1 for 24, 36, or 48 h or for 48 h at an MOI = 0.1, 0.5, or 1. *IL-1β* and *GAPDH* mRNAs were quantified by RT-PCR (**d, f**). IL-1β levels were determined by ELISA (**e** and **g**). **h–m** THP-1 macrophages were treated with 2 μM Nigericin for 2 h, infected with ZIKV at an MOI = 1 for 24, 36, and 48 h or for 48 h at an MOI = 0.1, 0.5, and 1. *IL-1β* and *GAPDH* mRNAs were quantified by RT-PCR (**h, j**). IL-1β levels were determined by ELISA (**i, k**). Mature IL-1β (p17) and cleaved Casp-1 (p22/p20) in supernatants or pro-IL-1β and pro-Casp-1 in lysates were determined by western blot (**l, m**). **n, o** BMDCs prepared from C57BL/6 mice bone marrow were stimulated by LPS (1 μg/ml) for 6 h or 2 μM Nigericin for 30 min or infected with ZIKV for 48 h at an MOI = 0.1, 0.5, and 1. *Pro-IL-1β* and *GAPDH* mRNAs were quantified by RT-PCR (**n**). IL-1β levels in supernatants were determined by ELISA (**o**). The number of replicates is two (**e, g, i, k, o**). The number of replicates is three (**d, f, h, j, n**). Data shown are means ± s.e.m.; *$P < 0.05$, **$P < 0.01$, ***$P < 0.0001$. ns, no significance (one-way ANOVA with Tukey's post-hoc test)

differentiated macrophages, HeLa cells, and Vero cells (Fig. 2j–l), suggesting that ZIKV facilitates NLRP3 speck formation to activate the inflammasome. The oligomerization of ASC is a direct indicator of inflammasome activation[30]. We further examined the effect of ZIKV on ASC pyroptosome formation in THP-1-differentiated macrophages. ASC was diffusely distributed in the nucleus and cytoplasm uninfected macrophages but formed distinct small specks in infected macrophages (Fig. 2m), and ASC oligomerization was facilitated by Nigericin and ZIKV in infected macrophages (Fig. 2n), suggesting that ZIKV facilitates ASC oligomerization. Taken together, we demonstrate that ZIKV activates NLRP3 inflammasome by facilitating NLRP3 speck formation and ASC oligomerization to induce IL-1β secretion.

**ZIKV proteins are involved in NLRP3 inflammasome activation.** The effects of ZIKV replication, proteins, and genomic RNA on the regulation of NLRP3 inflammasome were evaluated. In PBMCs, IL-1β protein secretion and mRNA expression were induced by LPS and ZIKV but not by ultraviolet-inactivated (UV-inactivated) or heat-inactivated (heat-inactivated) ZIKV (Fig. 3a, b), and ZIKV RNA was detected in ZIKV-infected cells but not in cells treated with ATP or with UV- or heat-inactivated ZIKV (Fig. 3c). In THP-1-differentiated macrophages, IL-1β secretion was stimulated by Nigericin and ZIKV but not by UV- or heat-inactivated ZIKV (Fig. 3d), *IL-1β* mRNA was increased by ZIKV but not by Nigericin or UV- or heat-inactivated ZIKV (Fig. 3e), ZIKV RNA was detected in infected cells but not in cells treated with Nigericin or inoculated with UV- or heat-inactivated

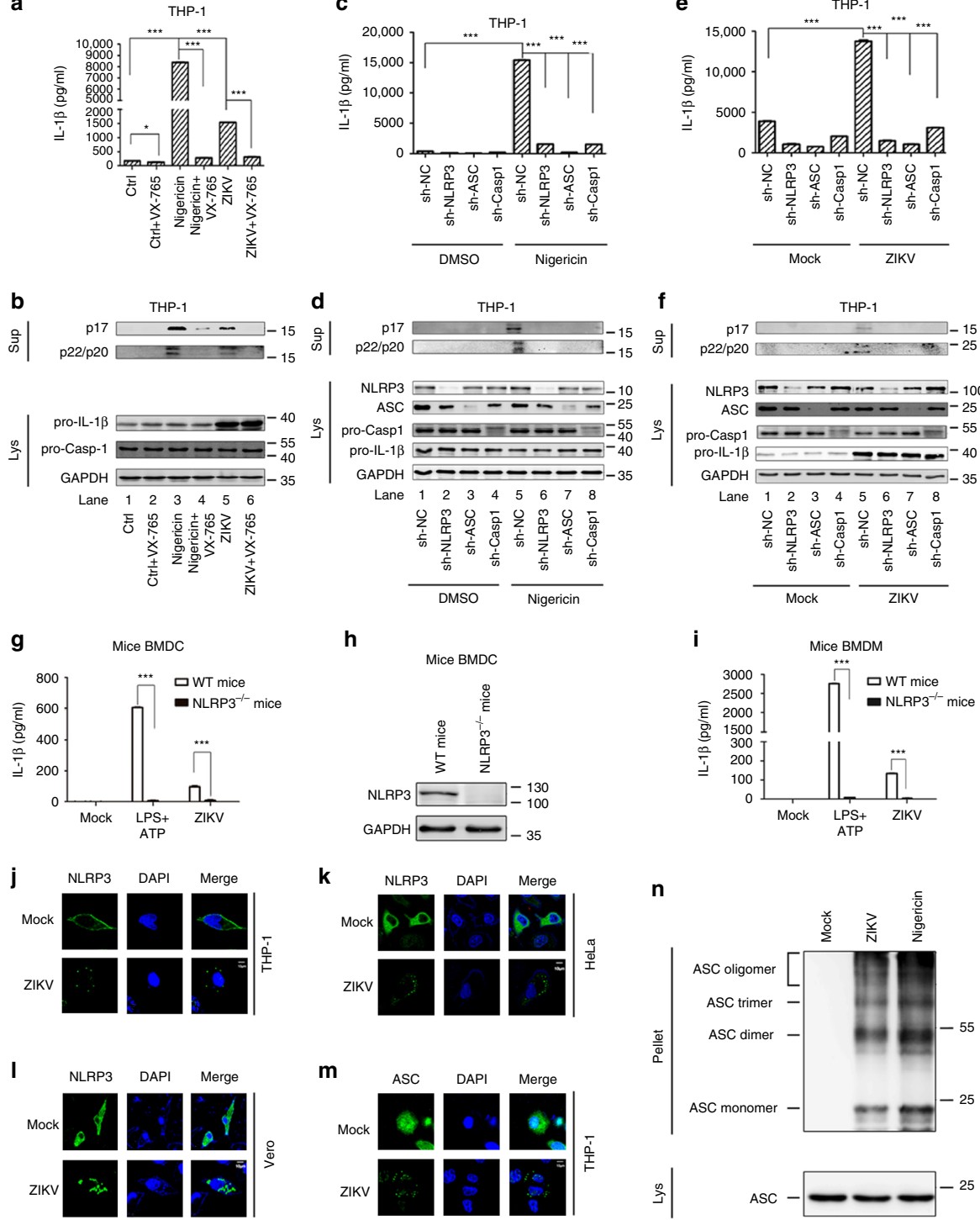

**Fig. 2** The role of NLRP3 inflammasome in regulation of ZIKV-induced IL-1β secretion. **a, b** THP-1 macrophages were treated with Casp-1 inhibitor VX-765 for 1 h or 2 μM Nigericin for 2 h or infected with ZIKV at an MOI = 1 for 24 h. **c–f** THP-1 cells stably expressing shRNAs targeting *NLRP3*, *ASC*, or *Casp-1* were generated and treated with 2 μM Nigericin for 2 h (**c, d**) or infected with ZIKV at an MOI = 1 for 24 h (**e** and **f**). IL-1β levels in the supernatants were determined by ELISA (**a, c, e**). p17 and p22/p20 levels in the supernatants were determined by western blot (**b, d, f**, top). NLRP3, ASC, pro-Casp-1, pro-IL-1β, Casp-1, and GAPDH proteins in the lysates were determined by western blot (**b, d, f**, bottom). (**g–i**) BMDCs prepared from the bone marrow (**g, h**) or BMDMs prepared from bone marrow cells (**i**) of treated C57BL/6 WT mice and C57BL/6 *NLRP3⁻ᐟ⁻* mice were stimulated with LPS (1 μg/ml) for 6 h and 5 mM ATP for 30 min or infected with ZIKV for 24 h at an MOI = 1. IL-1β levels in supernatants (**g, i**) and NLRP 3 proteins in lysates (**h**) were determined by western blot. (**j–m**) THP-1 macrophages were infected with ZIKV at an MOI = 1 for 24 h (**j, m**). HeLa cell line (**k**) and Vero cell line (**l**) that stably expressed NLRP3 were generated and infected with ZIKV at an MOI = 1 for 24 h. Subcellular localizations of NLRP3 (green) and ACS (green) and the nucleus marker DAPI (blue) were examined under confocal microscopy. Scale bar is 10 μm. **n** THP-1 macrophages were infected with ZIKV at an MOI = 1 for 24 h or treated with 2 μM Nigericin for 2 h. ASC oligomerization was determined by western blot using an anti-ASC antibody. The number of replicates is two (**a, c, e, g, i**). Data shown are means ± s.e.m; *P < 0.05, ***P < 0.0001 (one-way ANOVA with Tukey's post-hoc test)

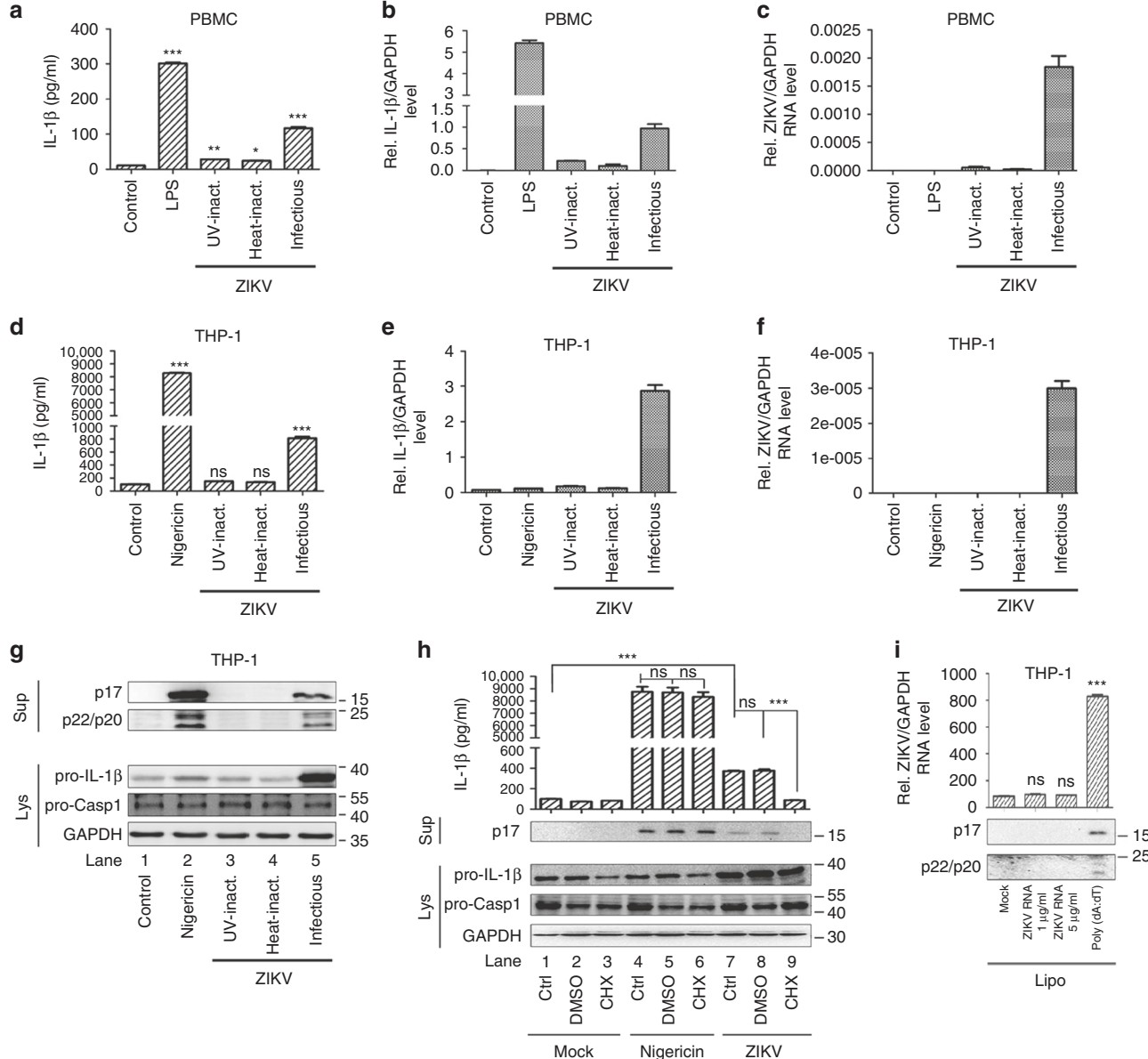

**Fig. 3** The requirements of ZIKV replication and protein production for the activation of the NLRP3 inflammasome. **a–g** Human PBMCs were treated with LPS (1 μg/ml) for 6 h (**a–c**) or THP-1-differentiated macrophages were treated with 2 μM Nigericin for 2 h (**d–g**) and then infected with ZIKV at an MOI = 1 for 48 h or inoculated with heat-inactivated or UV-inactivated ZIKV for 48 h. IL-1β levels in the supernatants were determined by ELISA (**a**, **d**). *Pro-IL-1β* and *GAPDH* mRNA (**b**, **e**) or ZIKV and *GAPDH* mRNA (**c**, **f**) levels were quantified by RT-PCR. Mature IL-1β (p17) and cleaved Casp-1 (p22/p20) levels in the supernatants (top) and pro-IL-1β, pro-Casp-1, and GAPDH levels in cell lysates (Lys) (bottom) were determined by western blot (**g**). **h** THP-1-differentiated macrophages were treated with 100 μM CHX for 1 h, treated with 2 μM Nigericin for 2 h or infected with ZIKV at an MOI = 1 for 24 h. IL-1β levels in the supernatants were determined by ELISA (top). Mature IL-1β (p17) levels in the supernatants and pro-IL-1β, pro-Casp-1, and GAPDH levels in cell lysates (Lys) were determined by western blot (bottom). **i** THP-1-differentiated macrophages were treated with Lipo, stimulated with Lipo plus poly(dA:dT), or treated with Lipo and transfected with 1 or 5 μg/ml genomic RNA of ZIKV for 6 h. IL-1β levels in the supernatants were determined by ELISA (top), and mature IL-1β and cleaved pro-Casp-1 protein levels in supernatants were determined by western blot (bottom). The number of replicates is two (**a**, **d**, **h**, **i**). The number of replicates is three (**b**, **c**, **e**, **f**). Data shown are means ± s.e.m; *$P < 0.05$, **$P < 0.01$, ***$P < 0.0001$; ns, no significance (one-way ANOVA with Tukey's post-hoc test)

ZIKV (Fig. 3f), and IL-1β and Casp-1 cleavages were activated by Nigericin and ZIKV but not by UV- or heat-inactivated ZIKV (Fig. 3g). These results suggest that ZIKV replication is required for the activation of IL-1β.

In addition, in THP-1-differentiated macrophages, in the absence of cycloheximide (CHX, a translation inhibitor)[31], IL-1β secretion, IL-1β maturation, and Casp-1 cleavage were induced by Nigericin and ZIKV, whereas in the presence of CHX, IL-1β secretion, IL-1β maturation, and Casp-1 cleavage were induced by Nigericin but not by ZIKV (Fig. 3h). These findings indicate that ZIKV protein production is required for IL-1β activation. Moreover, IL-1β secretion and Casp-1 cleavage were stimulated by poly(dA:dT) (an inflammasome activator)[32] but not by ZIKV genomic RNA (Fig. 3i). These results suggest that, unlike the genomic RNAs of influenza A virus (IAV) and hepatitis C virus (HCV)[33,34], ZIKV genomic RNA does not activate NLRP3

inflammasome. The effects of ZIKV genomic RNA on the transcription of NF-κB target genes were determined in THP-1-differentiated macrophages. The *pro-IL-1β, TNF-α, IL-8*, and *MCP-1* mRNAs were induced by LPS, ZIKV genomic RNA, HCV genomic RNA, and ZIKV infection (Supplementary Fig. 3a–d). ZIKV RNA was confirmed in cells transfected with ZIKV genomic RNA or infected with ZIKV, and HCV RNA was detected in cells transfected with HCV genomic RNA (Supplementary Fig. 3e, f). Thus ZIKV genomic RNA induces the NF-κB pathway but not the NLRP3 inflammasome or interferon (IFN)-1β secretion. Taken together, we reveal that ZIKV infection and protein production, but not ZIKV genomic RNA, are required for the activation of NLRP3 inflammasome.

**ZIKV NS5 facilitates the activation of NLRP3 inflammasome**. Next, we assessed which ZIKV protein is required for NLRP3 activation. A Casp-1 activation and IL-1β cleavage cell system was established based on previous reports[29,32,35], in which HEK293T cells were co-transfected with plasmids encoding NLRP3, ASC, pro-Casp1, and pro-IL-1β. The cells were then transfected with plasmids encoding each of the six ZIKV proteins. IL-1β secretion was activated by NS5, but not by Env, Capsid, NS2A, NS2B, or NS4B (Fig. 4a), indicating that NS5 is involved in the induction of IL-1β secretion. IL-1β secretion, IL-1β maturation, and Casp-1 cleavage were induced by NS5 (Fig. 4b), confirming that NS5 plays an important role in the activation of NLRP3 inflammasome.

NLRP3 undergoes a conformational change to form an active inflammasome complex in association with ASC[36]. The association of NS5 with NLRP3 inflammasome was thus evaluated. NLRP3 interacted with ASC and NS5 enhanced NLRP3-mediated recruitment of ASC (Fig. 4c), indicating that NS5 facilitates NLRP3 inflammasome activation. ASC oligomerization is critical for Casp-1 activation and inflammasome function[37]. In the presence of ASC, the ASC oligomerization was formed and enhanced by NLRP3 but not affected by NS5; however, in the presence of both ASC and NLRP3, the ASC oligomerization was stimulated by NLRP3 and further enhanced by NS5 (Fig. 4d). This result suggests that NS5 facilitates ASC oligomerization through interacting with NLRP3.

To confirm the role of NS5 in the activation of NLRP3 inflammasome, macrophages were transfected with a lentivirus expressing NS5 to generate a cell line that stably expresses NS5 (Supplementary Fig. 4). IL-1β secretion and cleavage, as well as ASC oligomerization was stimulated by ATP[38] and further enhanced by NS5 (Fig. 4e–g). Moreover, the cells that stably expressed NS5 were treated with a Casp-1 inhibitor, Ac-YVAD-cmk. The secretion of IL-1β and the cleavages of IL-1β and Casp-1 were induced by NS5, but such activations were inhibited by Ac-YVAD-cmk (Fig. 4h, i). Therefore, we suggest that ZIKV NS5 facilitates the activation of IL-1β and NLRP3 inflammasome and that Casp-1 is required for this regulation.

**NS5 binds NLRP3 by interacting with NACHT and LRR domains**. The mechanism by which NS5 activates NLRP3 inflammasome was investigated. In HEK293T cells, NS5 was co-precipitated with NLRP3 but not with ASC or Casp-1 (Fig. 5a, b), suggesting that NS5 associates with NLRP3 but not ASC or Casp-1. NLRP3 contains several prototypic domains, including a PYRIN domain, a NACHT domain, and seven LRR domains[39]. Based on this structure, four plasmids expressing NLRP3, PYRIN, NACHT, and LRR were constructed (Supplementary Fig. 5a). In HEK293T cells, NS5 was interacted with NLRP3, NACHT, and LRR but not with PYRIN (Fig. 5c), revealing that NS5 interacts with NLRP3 through NACHT and LRR domains. Yeast two-

hybrid screens confirmed the interaction of NS5 with LRR domain (Supplementary Fig. 5b). Glutathione S-transferase (GST) pull-down assays showed that GST-NS5 was pulled down with NLRP3 (Fig. 5d), indicating that NS5 directly binds NLRP3. GST-LRR was pulled down with NS5 (Fig. 5e), suggesting that NS5 binds LRR domain. Taken together, we demonstrate that NS5 binds NLRP3 by interacting with the NACHT and LRR domains.

**NS5 interacts with NLRP3 to facilitate inflammasome assembly**. Since NS5 directly binds NLRP3, the effects of this interaction on NLRP3 inflammasome activation and assembly were determined in HEK293T and HeLa cells. In the presence of single protein, NS5 was mainly localized in the nucleus, whereas NLRP3, ASC, or Casp-1 was diffusely distributed in cytoplasm (Fig. 6a and Supplementary Fig. 6a). In the presence of two proteins, NS5 was co-localized with NLRP3 and translocated from the nucleus to cytoplasm to form specks, but NS5 was not co-localized with ASC or Casp-1 and failed to form specks (Fig. 6b and Supplementary Fig. 6b). The interactions of NS5 with the domains of NLRP3 were then evaluated in HEK293T and HeLa cells (Supplementary Fig. 6c–f). In the presence of individual protein, NS5 was mainly localized in the nucleus, and NLRP3, PYRIN, or LRR was diffusely distributed in the cytoplasm, and by contrast, NACHT was localized in both the nucleus and cytoplasm (Supplementary Fig. 6c, d). In the presence of two proteins, a large proportion of NS5 was co-localized with NLRP3, NACHT, or LRR in the cytoplasm to form specks; however, most NS5 was not co-localized with PYRIN, remained in the nucleus, and failed to form specks (Supplementary Fig. 6e, f). Taken together, we reveal that NLRP3, but not ASC or Casp-1, interacts with NS5 through the NACHT and LRR domains to influence NS5 subcellular distribution.

Next, the effects of NS5 on the formation of NLRP3 inflammasome complex were explored in HEK293T cells. In the presence of individual protein, GFP was localized in both nucleus and cytoplasm, NS5 was mainly localized in the nucleus, NLPR3 was diffusely distributed in the cytoplasm, and ASC was distributed in the cytoplasm to form small ring structures (Fig. 6c). When two proteins presented, NLRP3 was co-localized with NS5 to form speck structure, and ASC was not co-localized with NS5 and distributed in the cytoplasm to form small ring structures (Fig. 6d). Moreover, in the absence of NS5, NLRP3 and ASC were co-localized and distributed in the cytoplasm to form "ring-like" structures, but in the presence of NS5, the three proteins (NLRP3, ASC, and NS5) were co-localized and distributed in the cytoplasm to from "sphere-like" structures, in which NS5 (green) was located inside, NLRP3 (red) was located in the middle, and ASC (cyan) was located outside (Fig. 6e). Taken together, we demonstrate that NS5 facilitates the NLRP3 inflammasome complex assembly by directly binding NLRP3 to form sphere-like structures of NS5–NLRP3–ASC.

**ZIKV infection induces inflammatory responses in mice**. The biological effects of ZIKV infection on the induction of inflammatory responses were evaluated in C57BL/6 WT mice and A129 mice deficient in IFN-α/β receptors[25]. C57BL/6 WT mice were treated with Ac-YVAD-cmk (caspase-1 inhibitor) and infected with ZIKV as described previously[40]. The viral RNA was detected in the blood of infected mice and slightly increased in the presence of Ac-YVAD-cmk (Supplementary Fig. 7a). Interestingly, IL-1β protein was induced in the sera of ZIKV-infected C57BL/6 WT mice, but such induction was repressed by Ac-YVAD-cmk (Fig. 7a), suggesting that ZIKV activates IL-1β production and caspase-1 is involved in such activation. The body weights of mock-infected mice were slightly increased during the treatment,

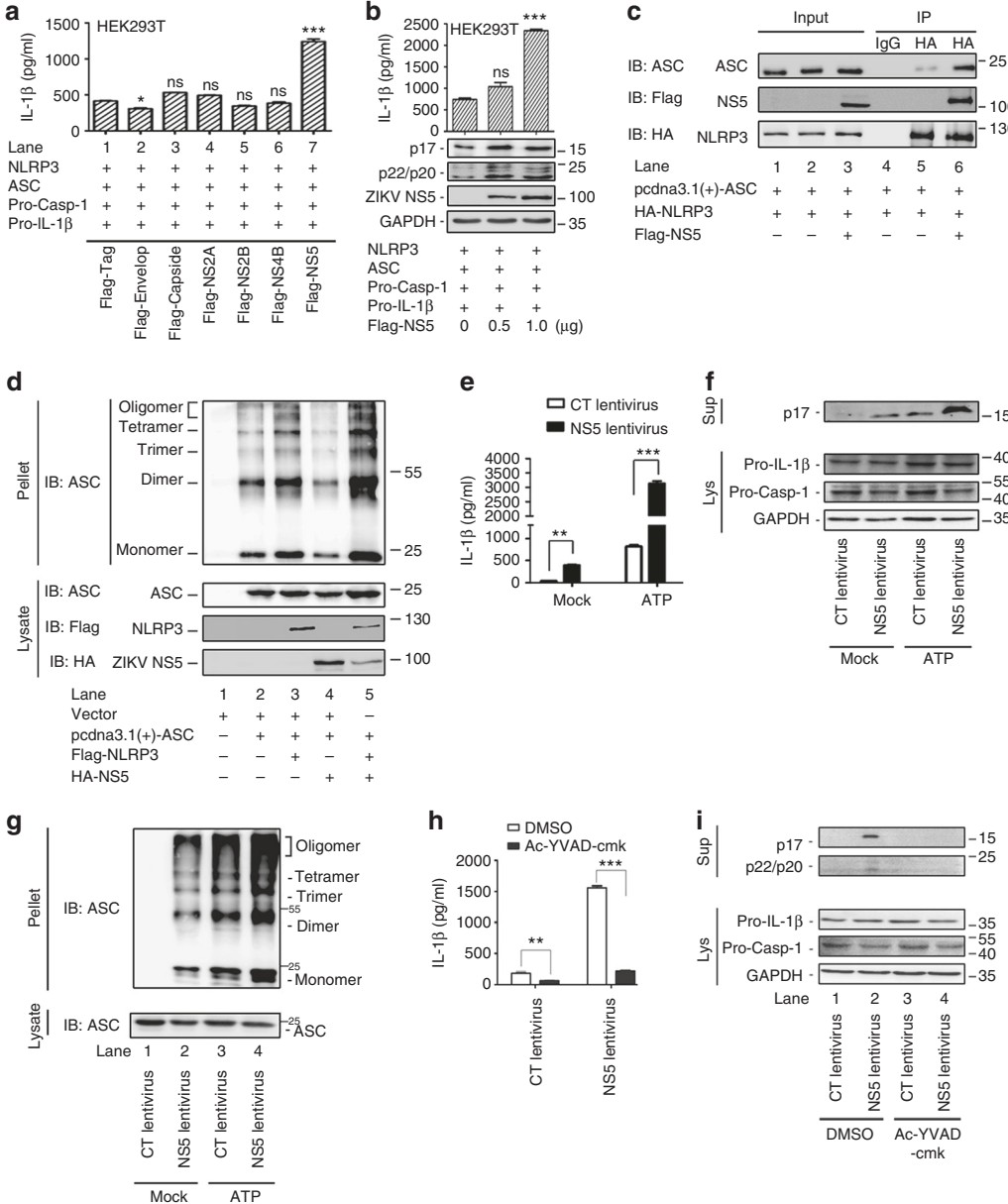

**Fig. 4** The role of ZIKV NS5 in the activation of the NLRP3 inflammasome. **a**, **b** HEK293T cells were co-transfected with plasmids encoding NLRP3, ASC, pro-Casp1, and pro-IL-1β. The cells were then transfected with pFlag-Env, pFlag-Capsid, pFlag-NS2A, pFlag-NS2B, pFlag-NS4B, or pFlag-NS5 (**a**) or transfected with pFlag-NS5 (**b**). IL-1β levels in the supernatants were determined by ELISA (**a**, **b**, top). The mature IL-1β, cleaved pro-Casp-1, ZIKV NS5, and GAPDH levels in the cell lysates were determined by western blot (**b**, bottom). **c** HEK293T cells were co-transfected with pcdna3.1(+)-ASC, pHA-NLRP3, or pFlag-NS5. Cell lysates were subjected to western blots (input) or IP using an IgG or anti-HA antibody and analyzed using immunoblotting with anti-HA, anti-ASC, or anti-Flag antibodies (IP). **d** HEK293T cells were co-transfected with pcdna3.1(+)-ASC, Flag-NLRP3 or HA-NS5 for 24 h. Cell lysates were prepared and the pellets were subjected to ASC oligomerization assays (top) or western blots (bottom). **e–g** THP-1 cells were infected with CT-lentivirus or NS5-lentivirus (NS5-encoding lentivirus), which were differentiated into macrophages by stimulating with TPA and then treated with 5 mM ATP for 2 h. IL-1β levels in the supernatants were determined by ELISA (**e**). p17 in the supernatants and pro-IL-1β, pro-Casp-1, and GAPDH in the cell lysates (Lys) were determined by western blot (**f**). Cell pellets were prepared and subjected to ASC oligomerization assays (**g**, top). ASC in the cell lysates was determined by western blot (**g**, bottom). **h**, **i** THP-1 cells were infected with CT-lentivirus or NS5-lentivirus, which were differentiated into macrophages by stimulating with TPA and then treated with caspase-1 inhibitor Ac-YVAD-cmk for 1 h. IL-1β levels in the supernatants were determined by ELISA (**h**). p17 in the supernatants and pro-IL-1β, pro-Casp-1, and GAPDH in the cell lysates (Lys) were determined by western blot (**i**). The number of replicates is two. Data shown are means ± s.e.m; *P < 0.05, **P < 0.01, ***P < 0.0001; ns, no significance (one-way ANOVA with Tukey's post-hoc test)

whereas ZIKV-infected mice started to lose weight during infection, but the reduction in body weight of ZIKV-infected mice was rescued by Ac-YVAD-cmk (Fig. 7b). Moreover, ZIKV infection induced splenomegaly and significant increased splenic index in the mice, but such inductions were repressed by Ac-YVAD-cmk (Fig. 7c). The viral RNA replication was confirmed in

the spleen of infected mice and slightly increased in the presence of YVAD-cmk (Supplementary Fig. 7b). Thus our results suggest that ZIKV infection may induce inflammatory responses in C57BL/6 WT mice.

To further confirm the biological effects of ZIKV infection on the induction of inflammatory responses, we used the A129 mice

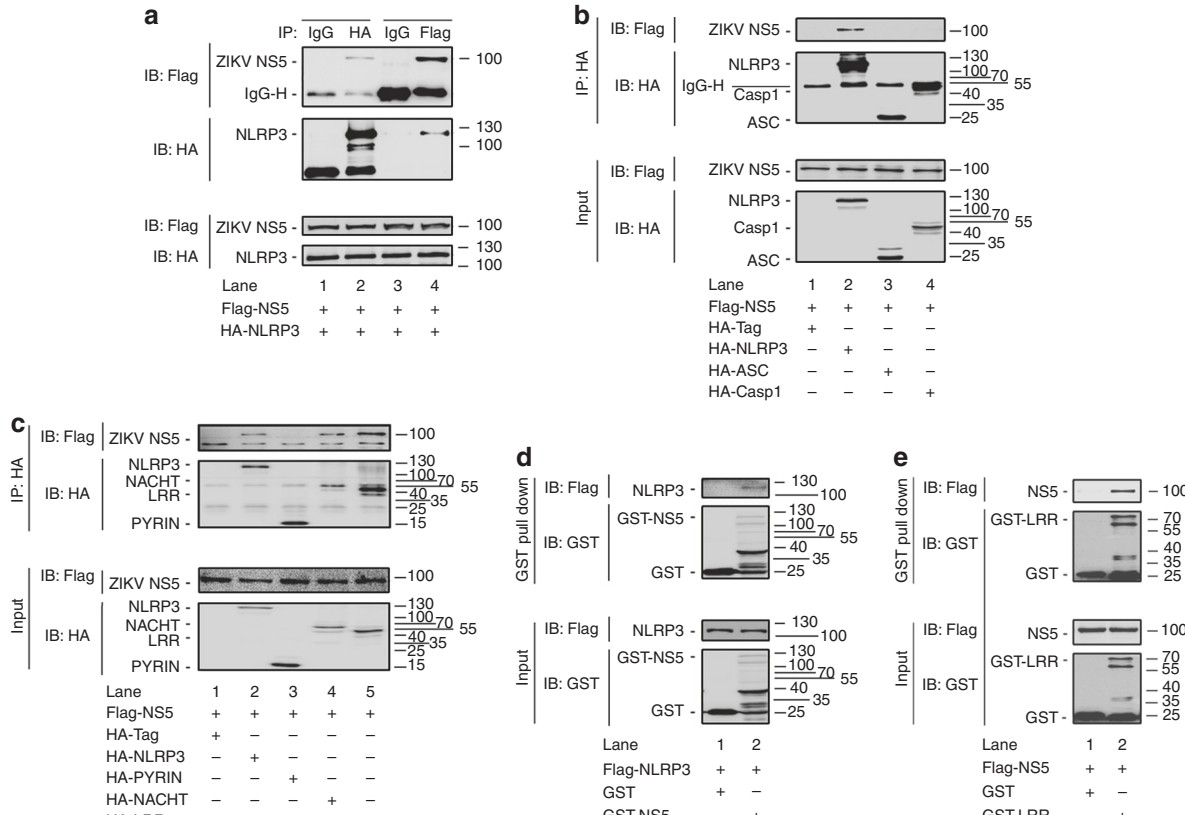

**Fig. 5** Interaction of ZIKV NS5 with NLRP3 protein. **a–c** HEK293T cells were co-transfected with pFlag-NS5 and pHA-NLRP3 (**a**); co-transfected with pFlag-NS5 and pHA-NLRP3, pHA-ASC, or pHA-Casp-1 (**b**); or co-transfected with pFlag-NS5 and pHA-NLRP3, pHA-PYRIN, pHA-NACHT, or pHA-LRR (**c**). Cell lysates were prepared and subjected to IP using a hepatitis C virus IgG, anti-HA, or anti-Flag antibody and analyzed by immunoblotting using an anti-HA or anti-Flag antibody (top) or subjected directly to western blot using an anti-Flag or anti-HA antibody (as input) (bottom). **d**, **e** HEK293T cells were transfected with pFlag-NLRP3 and incubated with 10 μg GST or GST-NS5 (**d**) or transfected with pFlag-NS5 and incubated with 10 μg GST or GST-LRR (**e**). Cell extracts were incubated with glutathione-Sepharose beads. Mixtures were analyzed by immunoblotting using an anti-Flag or anti-GST antibody (top). Cell lysates were analyzed by immunoblotting using an anti-Flag or anti-GST antibody (as input) (bottom)

model. The A129 mice model has been used by many research groups to study ZIKV infection[41] and pathogenesis, including microcephaly[42]. In the brain of infected mice, neutrophils and mononuclear cells were infiltrated and polymorphonuclear cells (PMNs) were detected near blood vessel, and in the spleen of infected mice, corpuscle structures were large and poorly defined (Fig. 7d). Dengue virus (DENV) activates the NLRP3 inflammasome in platelets to increase endothelium permeability[43]. Here we showed that ZIKV induced considerable vascular permeability in the infected mice liver and kidney (Fig. 7e). ZIKV E protein was not detected in the mock-infected mice brain, spleen, liver, or kidney but detected in the infected mice brain, spleen, liver, or kidney (Fig. 7f–i). These results demonstrate that ZIKV induces inflammatory responses in the mice brain and spleen. Moreover, the body weights of uninfected mice increased over time, but infected mice started to lose weight after 4 days postinfection (Fig. 7j). Finally, all uninfected mice survived beyond 12 days, infected mice began to die at 9 days postinfection, and all infected mice died after 12 days postinfection (Fig. 7k). Taken together, we reveal that ZIKV infection induces severe inflammatory responses in mice.

**ZIKV activates IL-1β to induce inflammatory responses**. The effect of ZIKV infection on the induction of inflammatory responses was further confirmed in mice. Initially, we evaluated

the role of IL-1β in ZIKV replication in A129 mice. At 6 days postinfection and in the blood of infected mice, the level of ZIKV titer ($3.7 \times 10^3$) was much lower in the absence of Ac-YVAD-cmk (Fig. 8a) as compared to that ($4.0 \times 10^5$) in the presence of Ac-YVAD-cmk (Fig. 8b), indicating that Caspase-1 may attenuate ZIKV replication. In addition, ZIKV E protein in the brain of infected mice was reduced by Ac-YVAD-cmk (Fig. 8c), but ZIKV E protein in the spleen, liver, or kidney of infected mice was enhanced by Ac-YVAD-cmk (Fig. 8d–f), demonstrating that Caspase-1 plays an important role in the regulation of ZIKV replication in mice.

Next, the effect of ZIKV infection on the activation of IL-1β was determined in A129 mice treated with Ac-YVAD-cmk. IL-1β protein secreted in the sera of infected mice were significantly induced by ZIKV, but such induction was repressed by Ac-YVAD-cmk (Fig. 8g), suggesting that Casp-1 is required for ZIKV-induced secretion of IL-1β. Similarly, IL-1β proteins produced in the brain, spleen, liver, and kidney of infected mice were activated by ZIKV, but such activations were inhibited by Ac-YVAD-cmk (Fig. 8h), indicating that Casp-1 is required for ZIKV-induced activation of IL-1β.

Moreover, the effect of ZIKV infection on the induction of inflammatory responses in mice was evaluated. Inflammatory responses in the brain, spleen, liver, and kidney of infected mice were induced by ZIKV, but such ZIKV-induced inflammatory responses were inhibited by Ac-YVAD-cmk

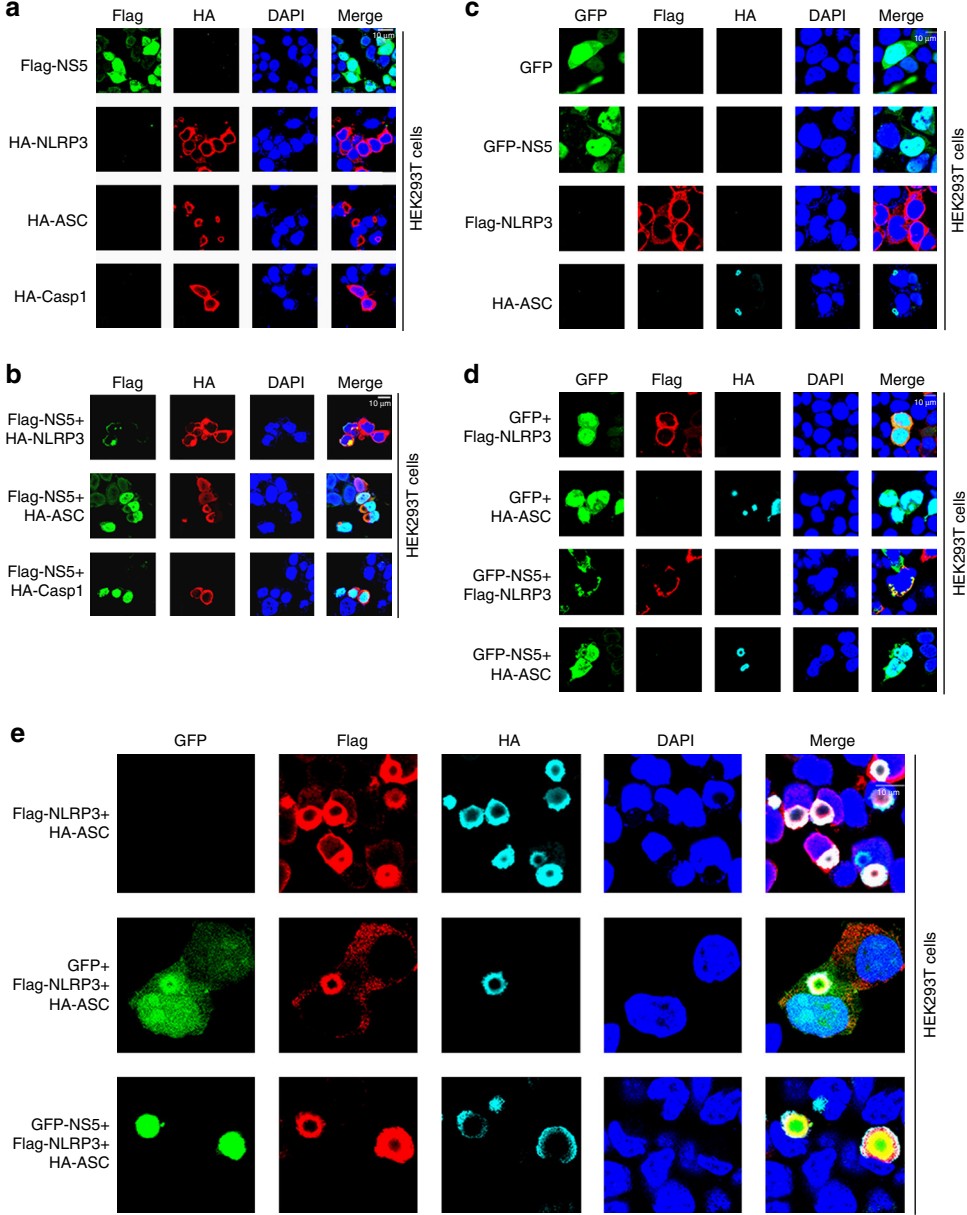

**Fig. 6** Role of ZIKV NS5 in NLRP3 inflammasome assembly. **a**, **b** HEK293T cells were transfected with pFlag-NS5, pHA-NLRP3, pHA-ASC, or pHA-Casp-1 (**a**) or co-transfected with pFlag-NS5 and pHA-NLRP3, pHA-ASC, or pHA-Casp-1 (**b**). Subcellular localizations of Flag-NS5 (green), HA-protein (red), and the nucleus marker DAPI (blue) were examined by confocal microscopy. **c-e** HEK293T cells were transfected with pGFP, pGFP-NS5, pFlag-NLRP3, or pHA-ASC (**c**); co-transfected with pGFP+pFlag-NLRP3, pGFP+pHA-ASC, pGFP-NS5+pHA-NLRP3, or pGFP-NS5+pHA-ASC (**d**); or co-transfected with pFlag-NLRP3+pHA-ASC, pGFP+pFlag-NLRP3+pHA-ASC, or pGFP-NS5+pFlag-NLRP3+pHA-ASC (**e**). Subcellular localizations of Flag-NS5 (green), HA-protein (red), and the nucleus marker DAPI (blue) were examined by confocal microscopy. Scale bar is 10 μm

(Fig. 8i), revealing that ZIKV induces inflammatory responses through Casp-1. Finally, the role of ZIKV infection in the growth of mice was evaluated. The body weights of infected mice were reduced during ZIKV infection, and such reductions were attenuated by Ac-YVAD-cmk (Fig. 8j), suggesting that Casp-1 or IL-1β is important for the infection of ZIKV. The survival rates of mice were reduced by ZIKV infection, and to our surprise, all infected mice survived in the presence of Ac-YVAD-cmk (Fig. 8k), further confirming that Casp-1 or IL-1β is important for the infection of ZIKV. Taken together, we demonstrate that ZIKV infection induces severe inflammatory responses in mice through Casp-1-mediated activation of IL-1β.

## Discussion

A fundamental component of the innate immune response of the host to viral infection is the production and release of the pro-inflammatory cytokine IL-1β[22]. This process is regulated by the NLRP3 inflammasome complex[39], in which NLRP3 together with ASC promotes the cleavage of pro-Casp-1, which regulates IL-1β maturation[20]. In this study, a novel mechanism by which ZIKV infection activates the NLRP3 inflammasome to facilitate IL-1β maturation is revealed (Fig. 9). ZIKV infection is associated with IL-1β secretion in infected patients, and ZIKV activates IL-1β secretion in infected human PBMCs, macrophages, mice, and mice BMDCs, suggesting that ZIKV infection activates IL-1β secretion. The activation of IL-1β secretion is regulated by two

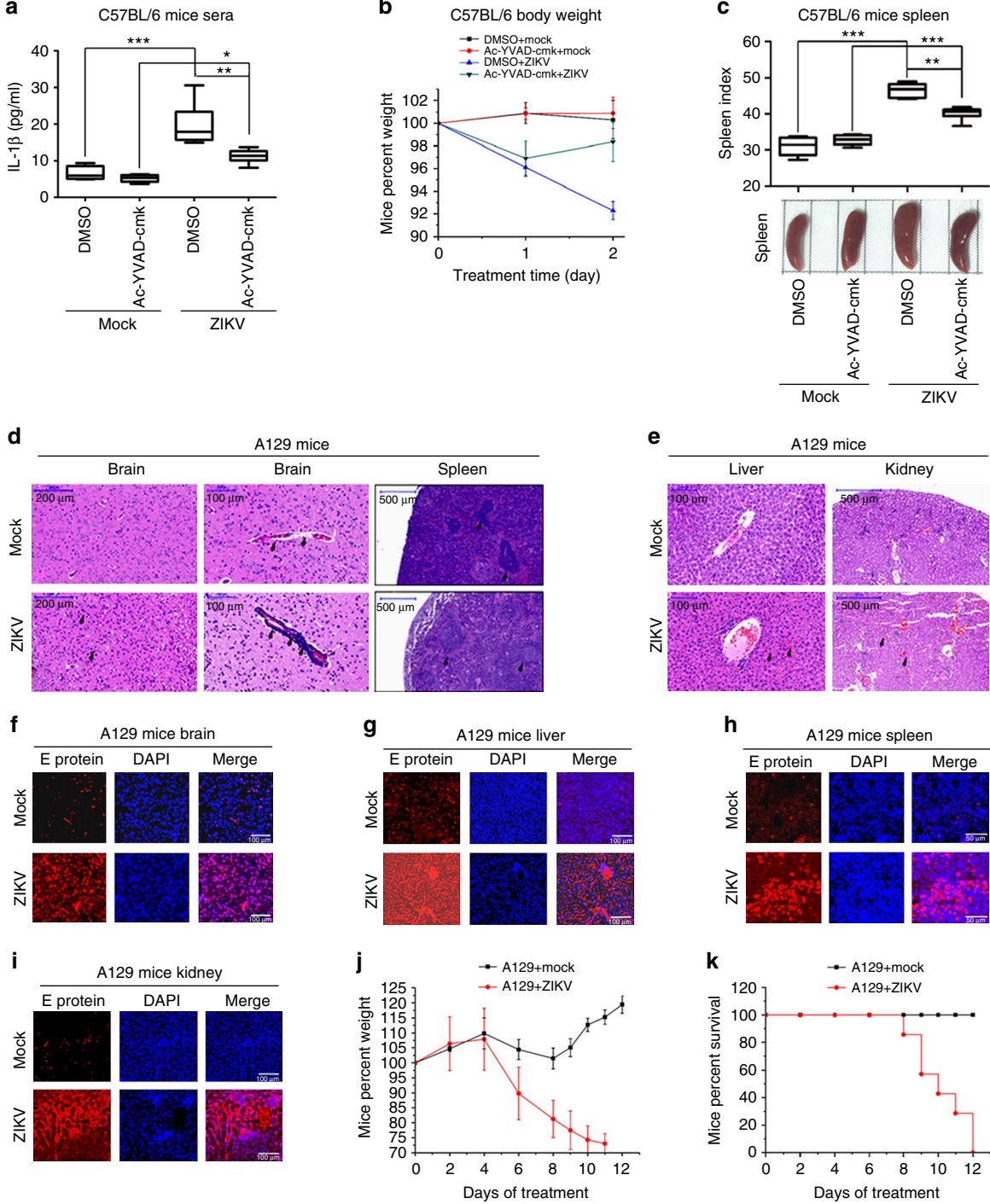

**Fig. 7** The effects of ZIKV infection on the induction of inflammatory responses in mice. **a–c** C57BL/6 WT mice (5 weeks old, male) were pretreated with DMSO ($n = 6$) or Ac-YVAD-cmk (8 mg/kg) ($n = 6$) by intraperitoneal injection for 30 mice and then infected with ZIKV ($5 \times 10^5$ PFU) ($n = 6$) or treated with PBS ($n = 6$) for 2 days. At 0, 1, and 2 days posttreatment, the levels of IL-1β protein in mice sera were determined by ELISA (**a**) and the mice body weights were evaluated (**b**). Data shown are whiskers: Min.–max.; \*$P < 0.05$, \*\*$P < 0.01$, \*\*\*$P < 0.0001$ (one-way ANOVA with Tukey's post-hoc test) (**a**). Data shown are means ± s.e.m (**b**). At 2 days posttreatment, the splenomegaly and splenic index (spleen weight (mg)/body weight (g) × 10) of treated mice was evaluated (**c**). Data shown are whiskers: Min.–max.; \*\*$P < 0.01$, \*\*\*$P < 0.0001$ (one-way ANOVA with Tukey's post-hoc test). **d–k** A129 mice (6 weeks old) ($n = 7$; 4 males and 3 females) were infected with ZIKV ($5 \times 10^5$ PFU) for the indicated times. A129 mice (6 weeks old) ($n = 3$; 2 males and 1 female) were treated with PBS as Mock-infection. **d–i** At 10 days posttreatment, mice brain tissue (**d**) and spleen tissue (**e**) were stained with hematoxylin and eosin (H&E) and examined using a light microscope, and the sections of mice brain tissue (**f**), spleen tissue (**g**), liver tissue (**h**), and kidney tissue (**i**) were stained with DAPI to label nuclei (blue) and an antibody against ZIKV E protein (red) and examined by confocal microscopy. Scale bar of the brain (left) is 200 μm. Scale bar of the brain (right) is 100 μm. Scale bar of the spleen is 500 μm (**d**). Scale bar of the liver is 100 μm. Scale bar of the kidney is 500 μm (**e**). Scale bar is 100 μm (**f–i**). (**j**, **k**) At 0, 2, 4, 6, 8, 10, and 12 days posttreatment, the mice body weights (**j**) and survival rates (**k**) were evaluated. Data shown are means ± s.e.m (**j**)

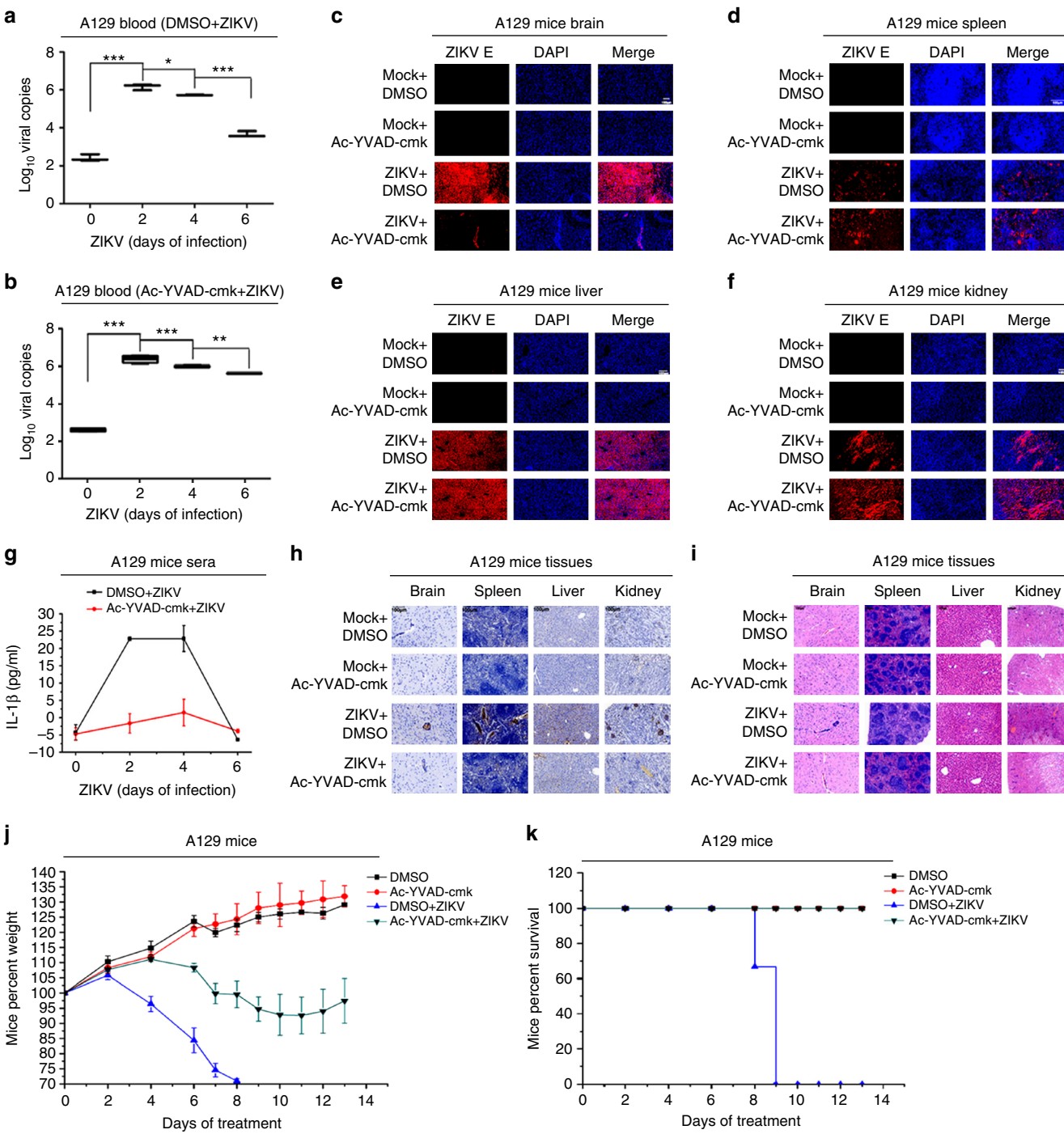

**Fig. 8** The effect of caspase-1 inhibitor on ZIKV-induced inflammatory responses in mice. A129 mice (6 weeks old) were treated with DMSO or Ac-YVAD-cmk (8 mg/kg) by intraperitoneal injection 30 min and infected with ZIKV ($5 \times 10^5$ PFU) or treated with PBS for the indicated times. **a, b** A129 mice were pretreated with DMSO ($n = 3$; 2 males and 1 female) (**a**) or Ac-YVAD-cmk ($n = 3$; 2 males and 1 female) (**b**) and then infected with ZIKV ($5 \times 10^5$ PFU) for 0, 2, 4, and 6 days. ZIKV RNA in the mice blood was determined by RT-PCR. Data shown are whiskers: Min.–max.; *$P < 0.05$, **$P < 0.01$, ***$P < 0.0001$ (one-way ANOVA with Tukey's post-hoc test). **c–f** A129 mice pretreated with DMSO ($n = 3$; 2 males and 1 female) or Ac-YVAD-cmk ($n = 3$; 2 males and 1 female) were infected with ZIKV ($5 \times 10^5$ PFU) for 11 days. The mice brain, spleen, liver, and kidney tissue sections were stained with DAPI to label nuclei (blue) and an antibody against ZIKV E protein (red) and then examined under confocal microscopy. Scale bar is 100 μm. **g** A129 mice pretreated with DMSO ($n = 3$; 2 males and 1 female) or Ac-YVAD-cmk ($n = 4$; 2 males and 2 females) were infected with ZIKV ($5 \times 10^5$ PFU) for 0, 2, 4, and 6 days. IL-1β levels in mice sera were determined by ELISA. Data shown are means ± s.e.m. **h–k** A129 mice pretreated with DMSO ($n = 3$; 2 males and 1 female) or Ac-YVAD-cmk ($n = 4$; 2 males and 2 females) were infected with ZIKV ($5 \times 10^5$ PFU). At 11 days posttreatment, the mice brain, spleen, liver, and kidney tissue samples were stained with IL-1β antibody (**h**) or stained with hematoxylin and eosin (H&E) (**i**) and then examined using a light microscope. At 0, 2, 4, 6, 7, 8, 9, 10, 11, 12, and 13 days posttreatment, the mice body weights (**j**) and survival rates (**k**) were evaluated. Scale bar is 100 μm (**h, i**). Data shown are means ± s.e.m (**j**)

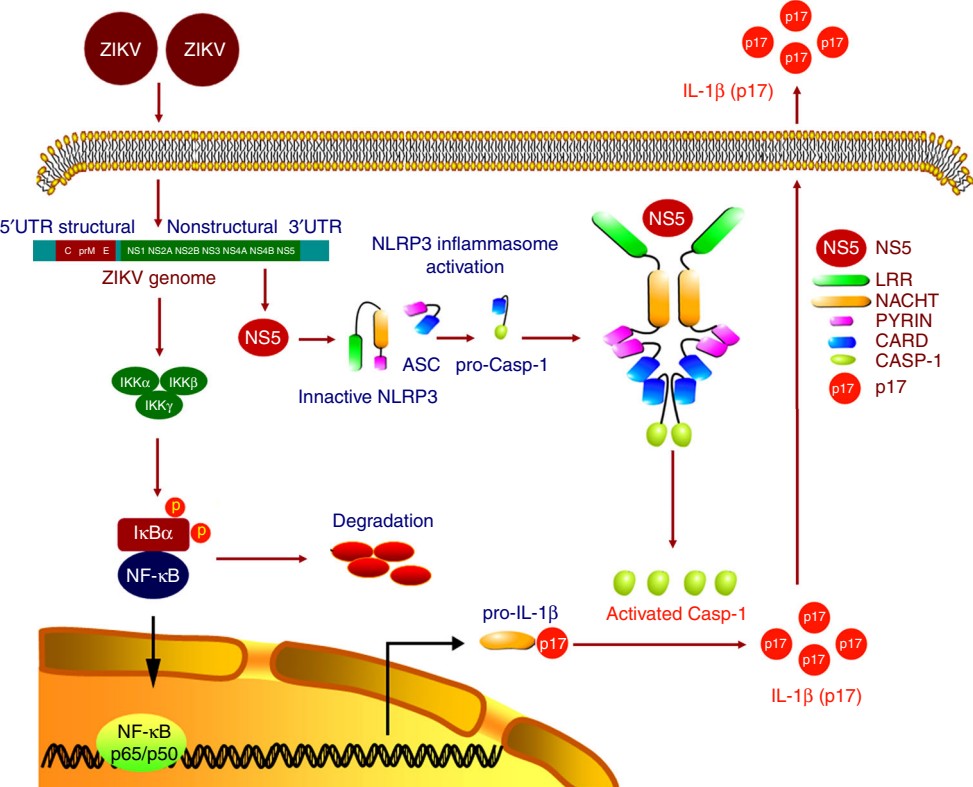

**Fig. 9** A proposed novel mechanism by which ZIKV infection induces inflammatory responses. During ZIKV infection, the viral NS5 protein activates the NLRP3 inflammasome by directly binding to NLRP3. The binding of NS5 to NLRP3 facilitates the assembly of NLRP3 inflammasome complex by forming a sphere-like structure of NS5-NLRP3-ASC. The activation of NLRP3 inflammasome promotes pro-Casp-1 cleavage, leading to the secretion of IL-1β. Then IL-1β plays crucial roles in host innate immunity by inducing host inflammatory responses

pathways in response to pathogens[19]. Knockdown of NLRP3 inflammasome components in THP-1-differentiated macrophages leads to the downregulation of ZIKV-induced secretion of IL-1β, demonstrating that the NLRP3 inflammasome is required for ZIKV-induced secretion of IL-1β. Knockout of *NLRP3* in C57BL/6 mice (*NLRP3*$^{-/-}$) results in the inhibition of ZIKV-induced IL-1β secretion, confirming that the NLRP3 inflammasome is essential for ZIKV-induced IL-1β secretion. The inflammasome complex is assembled by forming a ring-like structure[44], the localization of NLRP3 as speck is an indication of inflammasome complex formation[20], and ASC oligomerization is a direct indicator of inflammasome activation[30]. ZIKV facilitates NLRP3 speck formation and ASC oligomerization, suggesting that Casp-1 is required for ZIKV-induced IL-1β secretion. Thus we demonstrate that ZIKV infection triggers IL-1β secretion by activating the NLRP3 inflammasome.

To reveal the mechanism by which ZIKV activates the inflammasome, the effects of ZIKV replication, protein production, and genomic RNA were determined. *IL-1β* mRNA expression and protein secretion were activated by ZIKV but not by inactivated ZIKV, indicating that ZIKV infection is required for the NLRP3 inflammasome activation. ZIKV-induced IL-1β secretion and Casp-1 cleavage were abolished by the protein synthesis inhibitor CHX, demonstrating that protein production is essential for the activation of IL-1β. IL-1β secretion and Casp-1 cleavage were not affected by ZIKV genomic RNA, suggesting that ZIKV RNA is not involved in the NLRP3 inflammasome activation. This differs from previous reports that indicated that the NLRP3 inflammasome mediates innate immunity through recognition of IAV genomic RNA[33], and HCV genomic RNA activates the NLR3 inflammasome in human myeloid cells[34].

Thus we reveal that ZIKV infection and protein production are required for the NLRP3 inflammasome activation.

ZIKV encodes a single polypeptide that is proteolytically processed by viral and host proteases to produce three structural proteins and seven non-structural proteins[12]. We found that NS5 is required for regulation of the NLRP3 inflammasome. ZIKV NS5 is a viral RdRp responsible for the synthesis of viral genomic RNA through a de novo mechanism of initiation[13] and targets human signal transducer and activator of transcription factor 2 (STAT2) to inhibit type I IFN signaling[45]. Thus we reveal a novel function of NS5 in the regulation of NLRP3 inflammasome. The encephalomyocarditis virus viroporin 2B activates the NLRP3 inflammasome by inducing Ca$^{2+}$ flux[31]. We recently reported that enterovirus 71 3D protein enhances the NLRP3 inflammasome complex assembly[29]. Here we reveal a novel mechanism by which ZIKV NS5 activates the NLRP3 inflammasome. NS5 is mainly localized in the nucleus, which is consistent with a previous report[45]; by contrast, NLRP3, ASC, and Casp-1 are diffusely distributed in the cytosol. NS5 interacts with NLRP3 (but not ASC or Casp-1) through the NACHT and LRR domains of the latter in the cytosol. Interestingly, NS5 binds NLRP3 to facilitate NLRP3 inflammasome complex assembly by forming a sphere-like structure of NS5–NLRP3–ASC, in which NS5 locates inside, NLRP3 sites in the middle, and ASC distributes outside.

More importantly, the biological effects of ZIKV infection on the induction of inflammatory responses are evaluated in C57BL/6 wild-type (WT) mice and A129 mice deficient in IFN-α/β receptors. In C57BL/6 WT mice, ZIKV infection activates IL-1β production and induces inflammatory responses, and Casp-1 is involved in such activations. Moreover, in A129 mice, ZIKV infection significantly induces IL-1β production in the brain,

spleen, liver and kidney, but such inductions are significantly suppressed by the caspase-1 inhibitor (Ac-YVAD-cmk), confirming that Casp-1 is required for ZIKV-induced activation IL-1β. We further reveal that ZIKV infection induces severe inflammatory responses in A129 mice. Neutrophils and mononuclear cells infiltrated infected mice brains and PMNs were detected near blood vessels. In the spleens of infected mice, the spleen corpuscle structures were poorly defined. DENV activates the NLRP3 inflammasome in platelets to increase endothelium permeability[43]. Interestingly, ZIKV infection induces significant vascular permeability in the mime liver and kidney, and ZIKV-induced inflammatory responses are inhibited by Ac-YVAD-cmk, revealing that ZIKV induces inflammatory responses through Casp-1.

It has been reported that AG129 mice deficient in IFN α/β and γ receptors are highly susceptible to ZIKV infection and diseases[46]. In this study, ZIKV-infected A129 mice started to lose weight at 4 days postinfection and began to die at 9 days postinfection. ZIKV-induced reduction in mice body weights is attenuated by Ac-YVAD-cmk, and all infected mice survived in the presence of Ac-YVAD-cmk, demonstrating that ZIKV induces severe inflammatory responses in mice through Casp-1-mediated activation of IL-1β. Taken together, we demonstrate that ZIKV facilitates the NLRP3 inflammasome assembly and activation, which leads to activation of IL-1β secretion and induction of aggressive host inflammatory responses that are major determinants of ZIKV pathogenicity.

Mosquito-borne flaviviruses, including West Nile virus (WNV), DENV, and ZIKV[1,47], have caused considerable epidemics. NLRP3 is required for immune defense of host against pathogens, including WNV and DENV[43,48]. IL-1β is an important component of the host defense against viral infection, due to its key role in innate immunity. It is a highly potent pro-inflammatory mediator that induces vasodilation and attracts granulocytes to the inflamed tissues[49]. DENV infection triggers the NLRP3 inflammasome activation to increase vascular permeability by releasing IL-1β. This may be a major mechanism by which DENV induces severe dengue fever[50]. In addition to its function in innate immunity, IL-1β induces production of prostaglandin E2 in the hypothalamus, resetting the hypothalamic thermostat to fever[51]. Moreover, IL-1β is involved in the afferent pain response by signaling inflammatory pain to the hypothalamus[52], and the overproduction of IL-1β induces a number of genetically determined auto-inflammatory disorders[53]. These diseases usually manifest as fever, high acute-phase responses, arthritis, and rash, symptoms similar to ZIKV-associated illness. ZIKV evades IFN signaling, which is important for host antiviral innate immunity, through NS5-mediated degradation of STAT2[54]. We demonstrate that ZIKV NS5 induces IL-1β secretion by activating the NLRP3 inflammasome and thus reveal a novel function of NS5 in the regulation of host innate immunity. Thus ZIKV NS5 as an activator of the NLRP3 inflammasome plays important roles in ZIKV infection and associated diseases.

## Methods

**Clinical specimens and blood samples.** Sera of ZIKV-infected patients ($n = 11$) and healthy individuals ($n = 13$) were provided by the Institute of Pathogenic Microbiology, Center for Disease Control and Prevention of Guangdong (Guangzhou, China). Blood samples of healthy donors were randomly collected from the Wuhan Blood Donation Center (Wuhan, China). To isolate PBMCs, blood cells were separated from blood samples and diluted in RPMI-1640 purchased from Gibco (Grand Island, USA). Diluted blood cells (5 ml were added gently to a 15 ml centrifuge tube containing 5 ml lymphocyte separation medium (#50494) purchased from MP Biomedicals (Santa Ana, USA) and centrifuged at $2000 \times g$ for 10 min at room temperature (RT). The middle layer was transferred to a new centrifuge tube and diluted with RPMI-1640. The remaining red blood cells were removed using a red blood cell lysis buffer (Sigma-Aldrich, St. Louis, USA).

The purified PBMCs were centrifuged at $1500 \times g$ for 10 min at RT and cultured in RPMI-1640.

The study was conducted according to the principles of the Declaration of Helsinki and approved by the Institutional Review Board of the College of Life Sciences, Wuhan University in accordance with its guidelines for the protection of human subjects. The Institutional Review Board of the College of Life Sciences, Wuhan University approved the collection of blood samples for this study, and it was conducted in accordance with the guidelines for the protection of human subjects. Written informed consent was obtained from each participant.

**Animal study.** C57BL/6 WT mice were purchased from Hubei Research Center of Laboratory Animals (Wuhan, Hubei, China). C57BL/6 $NLRP3^{-/-}$ mice were kindly provided by Dr. Di Wang of Zhejiang University School of Medicine, China. 129/Sv/Ev mice (A129 mice) deficient in IFN-α/β receptors[25] were kindly provided by Dr. Cheng-Feng Qin of the Beijing Institute of Microbiology and Epidemiology, China.

Mouse BMDCs were differentiated from fresh bone marrow cells of C57BL/6 WT mice and C57BL/6 $NLRP3^{-/-}$ mice in RPMI-1640 medium containing 10% heat-inactivated fetal bovine serum (FBS) in the presence of granulocyte macrophage colony-stimulating factor in six-well plates for 5 days. The culture medium was replaced every other day.

Mouse BMDMs were differentiated from fresh bone marrow cells of C57BL/6 WT mice and $NLRP3^{-/-}$ mice. The bone marrow cells were incubated in six-well plates for 10 days with 10% L929-conditioned, 10% heat-inactivated FBS in RPMI-1640 medium. The culture medium was replaced every 2 days.

The animal study was approved by the Institutional Review Board of the College of Life Sciences, Wuhan University and was conducted in accordance with the guidelines for the protection of animal subjects.

**Cell lines and cultures.** African green monkey kidney epithelial cells (Vero) (ATCC, #CCL-81), *Aedes albopictus* mosquito cells (C6/36) (ATCC, #CRL-1660), and human embryonic kidney cells (HEK293T) were purchased from the American Type Culture Collection (ATCC, #CRL-3216) (Manassas, VA, USA). The human monocytic cell line THP-1 was a gift from Dr. Bing Sun of the Institute of Biochemistry and Cell Biology, Shanghai Institute for Biological Sciences, China. THP-1 cells were cultured in RPMI-1640 medium supplemented with 10% heat-inactivated FBS, 100 U/ml penicillin, and 100 μg/ml streptomycin sulfate. Vero and HEK293T cells were cultured in Dulbecco's modified Eagle's medium (DMEM) (Gibco, Grand Island, NY, USA) supplemented with 10% FBS, 100 U/ml penicillin, and 100 μg/ml streptomycin sulfate. Vero, HEK293T, and THP-1 cells were maintained in an incubator at 37 °C in a humidified atmosphere of 5% $CO_2$. C6/36 cells were maintained in an incubator at 30 °C in a humidified atmosphere of 5% $CO_2$.

The free of mycoplasma contamination of Vero and C6/36 cell stocks used in this study was tested by using the MycoTest Kit (ChanGEnome, Chian) and by using transmission electron microscope (TEM).

**Reagents.** LPS, ATP, TPA, and dansylsarcosine piperidinium salt (DSS) were purchased from Sigma-Aldrich (St. Louis, MO, USA). RPMI-1640 and DMEM were obtained from Gibco (Grand Island, NY, USA). Nigericin and Ac-YVAD-cmk were obtained from InvivoGene Biotech Co., Ltd. (San Diego, CA, USA). The caspase-1 inhibitor VX-765 was purchased from Selleckchem (Houston, TX, USA). Antibody against Flag (F3165) (1:2000) and monoclonal mouse anti-GAPDH (G9295) (1:5000) were purchased from Sigma (St Louis, MO, USA). Monoclonal rabbit anti-NLRP3 (D2P5E) (1:1000), monoclonal rabbit anti-IL-1β (D3U3E) (1:1000), IL-1β mouse mAb (3A6) (1:1000), and monoclonal rabbit anti-caspase-1 (catalog no. 2225) (1:1000) were purchased from Cell Signaling Technology (Beverly, MA, USA). Monoclonal mouse anti-ASC (sc-271054) (1:500) and polyclonal rabbit anti-IL-1β (sc-7884) (1:500) were purchased from Santa Cruz Biotechnology (Santa Cruz, CA, USA). Monoclonal mouse anti-NLRP3 (ALX-804-818) (1:500) was purchased from Enzo Life Sciences (Shanghai, China) to detect endogenous NLRP3 in THP1 cells by immunofluorescence microscopy. Lipofectamine 2000, normal rabbit immunoglobulin G (IgG), and normal mouse IgG were purchased from Invitrogen Corporation (Carlsbad, CA, USA).

**Viruses.** The ZIKV isolate z16006 (GenBank accession number, KU955589.1) isolated by the Institute of Pathogenic Microbiology, Center for Disease Control and Prevention of Guangdong (Guangzhou, Guangdong, China) was used in this study. C6/36 cells were maintained at 30 °C in DMEM (Gibco) (Grand Island, NY, USA) supplemented with 10% heat-inactivated FBS with penicillin and streptomycin (Gibco) (Grand Island, NY, USA) and 1% tryptose phosphate broth (Sigma) (St Louis, MO, USA). HCV genotype 2a strain JFH-1 was kindly provided by Dr. Takaji Wakita of the National Institute of Infectious Diseases, Japan.

To free the ZIKV stocks used in this study of mycoplasma contamination, it was tested by using the MycoTest Kit (ChanGEnome, Chian) and by using TEM.

**ZIKV RNA isolation and viral titer determination.** Vero cells were infected with ZIKV at an infection of 0.1 plaque-forming unit (PFU) per cell. The supernatant was collected when cells showed maximal cytopathic effect from viral infection,

centrifuged at $1100 \times g$ for 30 min, and then passed through 0.4 μm filters. We used the E.Z.N.Z. viral RNA Kit purchased from Omega Bio-Tek (Norcross, GA, USA) for the isolation of ZIKV RNA from the cell culture supernatant. The concentration of viral RNA was measured by NanoDrop 2000 purchased from Thermo Fisher Scientific (Waltham, MA, USA).

The standard curves were generated by the infectious cDNA clone of ZIKV, which was kindly provided by Dr. Pei-Yong Shi of the University of Texas Medical Branch, USA. A series of 10-fold dilution equivalent to $1 \times 10^2$–$1 \times 10^7$ copies per reaction mixture was prepared in parallel with the test samples.

**Plasmid construction**. The cDNAs encoding human *NLRP3*, *ASC*, *pro-Casp-1*, and *IL-1β* were obtained by reverse transcription of total RNA from TPA-differentiated THP-1 cells, followed by PCR using specific primers. The cDNAs were subcloned into pcDNA3.1(+) and pcagg-HA vector. The pcDNA3.1(+)-3 × Flag vector was constructed from pcDNA3.1(+) vector through inserting the 3 × Flag sequence between the *Nhe*I and *Hind*III site. The primers used in this study are shown in Supplementary Table 1.

To construct plasmids expressing ZIKV Envelop, Capsid, NS2A, NS2B, NS4B, and NS5 protein corresponding fragments of ZIKV cDNA were cloned into pcDNA3.1(+)-3 × Flag vector. To construct pGEX6p1-NS5, the ZIKV NS5 region was subcloned into pGEX6p-1 vector using *Eco*RI and *Xho*I sites. To construct pGEX6p1-LRR, the *LRR* region was subcloned into pGEX6p-1 vector using *Bam*HI and *Xho*I sites. The PYRIN, NACHT, and LRR domain of NLRP3 protein was cloned into pcaggs-HA vector using specific primers shown in Supplementary Table 1.

*Lentivirus production and infection*: The targeting sequences of shRNAs for the human *NLRP3*, *ASC*, and *Casp-1* were as follows: sh-NLRP3: 5′-CAGGTTTGACTATCTGTTCT-3′; sh-ASC: 5′-GATGCGGAAGCTCTTCAGTTTCA-3′; and sh-caspase-1: 5′-GTGAAGAGATCCTTCTGTA-3′. A PLKO.1 vector encoding shRNA for a negative control (Sigma-Aldrich, St. Louis, MO, USA) or a specific target molecule (Sigma-Aldrich) was transfected into HEK293T cells together with psPAX2 and pMD2.G with Lipofectamine 2000. We using the 3 × Flag sequence to replace the GFP protein in the pLenti CMV GFP Puro vector (Addgene, 658-5) for adding some Restriction Enzyme cutting site (*Xba*I-*Eco*RV-*Bst*BI-*Bam*HI) before the 3 × Flag tag. Then the pLenti vector encoding ZIKV NS5 protein was transfected into HEK293T cells together with psPAX2 and pMD2.G with Lipofectamine 2000. The primers are shown in Supplementary Table 1. Culture supernatants were harvested 36 and 60 h after transfection and then centrifuged at $1000 \times g$ for 15 min. THP-1 cells were infected with the supernatants containing lentiviral particles in the presence of 4 μg/ml polybrene (Sigma). After 48 h of culture, cells were selected by 1.5 μg/ml puromycin (Sigma). The results of each sh-RNA-targeted protein and the lenti-NS5 protein were detected by real-time PCR or immunoblot analysis.

**Enzyme-linked immunosorbent assay (ELISA)**. The concentrations of IL-1β in culture supernatants were measured by the ELISA Kit (BD Biosciences, San Jose, CA, USA). The mouse IL-1β ELISA Kit was purchased from the 4A Biotech Co. Ltd. (Beijing, China).

**Activated caspase-1 and mature IL-1β measurement**. The supernatant (1 ml) of cultured cells was collected in cryogenic vials (Corning) and stored frozen at −80 °C for 4 h. A rotational vacuum concentrator (Martin Christ) was used to lyophilize the samples. The lyophilized product was dissolved in 100 μl phosphate-buffered saline (PBS) and mixed with sodium dodecyl sulfate (SDS) loading buffer for western blot analyses using antibodies against activated caspase-1 (D5782 1:500; Cell Signaling) or mature IL-1β (Asp116 1:500; Cell Signaling). Adherent cells were lysed with the lysis buffer described below, followed by immunoblot analyses of various proteins. The uncropped scans of western blots are shown in Supplementary Fig. 8.

**Plaque assay**. Vero cells were cultured in a 12-well plate at a density of $2 \times 10^5$ cells/well and infected with 100 μl serially diluted cell culture supernatant for 2 h. Then the cells were washed by PBS and then immediately replenished with plaque medium supplemented with 1% carboxylmethylcelluose. The infected Vero cells were incubated for 5 days. After incubation, plaque medium was removed and cells were fixed and stained with 4% formaldehyde and 0.5% crystal violet.

**THP-1 macrophage stimulation**. THP-1 cells were differentiated to macrophages with 60 nM TPA for 12–14 h, and cells were cultured for 24 h without TPA. And then the differentiated cells were stimulated in 6 cm plates with ZIKV, LPS, Nigericin, or ATP. Supernatants were collected for the measurement of IL-1β by ELISA. Cells were harvested for real-time PCR or immunoblot analysis.

**Activated caspase-1 and mature IL-1β measurement**. The supernatant (1 ml) of the cultured cells was collected in the cryogenic vials (Corning). The supernatant was frozen in −80 °C for 4 h. The Rotational Vacuum concentrator machine, which was purchased from Martin Christ, was used for freeze drying. The drying product was dissolved in 100 μl PBS and mixed with SDS-loading buffer for western

blotting analysis with antibodies for detection of activated caspase-1 (D5782 1:500; Cell Signaling) or mature IL-1β (Asp116 1:500; Cell Signaling). Adherent cells in each well were lysed with the lyses buffer described below, followed by immunoblot analysis to determine the cellular content of various protein.

**Western blot analysis**. HEK293T whole-cell lysates were prepared by lysing cells with buffer (50 mM Tris-HCl, pH7.5, 300 mM NaCl, 1% Triton-X, 5 mM EDTA, and 10% glycerol). The TPA-differentiated THP-1 cells lysates were prepared by lysing cells with buffer (50 mM Tris-HCl, pH7.5, 150 mM NaCl, 0.1% Nonidetp 40, 5 mM EDTA, and 10% glycerol). Protein concentration was determined by Bradford assay (Bio-Rad, Hercules, CA, USA). Cultured cell lysates (30 μg) were electrophoresed in an 8–12% SDS-polyacrylamide gel electrophoresis (PAGE) gel and transferred to a polyvinylidene difluoride (PVDF) membrane (Millipore, MA, USA). PVDF membranes were blocked with 5% skim milk in PBS with 0.1% Tween 20 (PBST) before being incubated with the antibody. Protein band were detected using a Luminescent image Analyzer (Fujifilm LAS-4000).

**Co-immunoprecipitation assays**. HEK293T whole-cell lysates were prepared by lysing cells with buffer (50 mM Tris-HCl, pH7.5, 300 mM Nacl, 1% Triton-X, 5 mM EDTA, and 10% glycerol). TPA-differentiated THP-1 cell lysates were prepared by lysing cells with buffer (50 mM Tris-HCl, pH7.5, 150 mM Nacl, 0.1% Nonidetp40, 5 mM EDTA, and 10% glycerol). Lysates were immunoprecipitated with control mouse IgG (Invitrogen) or anti-Flag antibody (Sigma, F3165) (1:2000) with Protein-G Sepharose (GE Healthcare, Milwaukee, WI, USA).

**Confocal microscopy**. TPA-differentiated THP-1 cells was cultured or infected by the ZIKV (multiplicity of infection = 1) for 24 h. Cells were fixed in 4% paraformaldehyde at RT for 15 min. After washing three times with PBS, the cells were permeabilized with PBS containing 0.1% Triton X-100 for 5 min, washed three times with PBS, and finally blocked with PBS containing 5% BSA for 1 h. The cells were then incubated with the monoclonal mouse IgG1 anti-NLRP3 antibody (ALX-804-818-C100; Enzo life Sciences) (1:500) overnight at 4 °C, followed by incubation with fluorescein isothiocyanate-conjugated donkey anti-mouse IgG for 1 h. After washing three times, cells were incubated with 4,6-diamidino-2-phenylindole solution for 5 min and then washed three more times with PBS. Finally, the cells were analyzed using a confocal laser scanning microscope (Fluo View FV1000; Olympus, Tokyo, Japan).

**Real-time PCR**. Total RNA was extracted with TRIzol reagent (Invitrogen) following the manufacturer's instructions. Real-time quantitative reverse transcriptase-PCR (RT-PCR) was performed using the Roche LC480 and SYBR RT-PCR Kits (DBI Bio-science, Ludwigshafen, Germany) in a reaction mixture of 20 μl SYBR Green PCR master mix, 1 μl DNA diluted template, and RNase-free water to complete the 20 μl volume. Real-time PCR primers were designed by Primer Premier 5.0 and their sequences are provided in Supplementary Table 2.

**GST pull-down assays**. The plasmids pGEX6p1-NS5 and pGEX6p1-LRR were transfected into *Escherichia coli* strain BL21. After growing in LB medium at 37 °C until the OD600 reached 0.6–0.8, isopropyl β-D-1-thiogalactopyranoside was added to a final concentration of 1 mM, and the cultures were grown for an additional 4 h at 37 °C for GST-NS5 and GST-LRR protein. And then the GST protein, GST-NS5 protein, and GST-LRR protein were purified from *E. coli* bacteria. For GST-NS5 pull-down assay, glutathione-Sepharose beads (Novagen) were incubated with GST-NS5 or GST protein. After washing with PBS, these beads were incubated with cell lysates from HEK293T, which were transfected with plasmids encoding Flag-NLRP3 for 4 h at 4 °C. The precipitates were washed three times, boiled in 2× SDS-loading buffer, separated by 10% SDS-PAGE, and immunoblotted with anti-GST, anti-Flag. It was the same for the GST-LRR pull-down assay.

**Yeast two-hybrid analyses**. *Saccharomyces cerevisiae* strain AH109 and control vectors pGADT7, pGBKT7, pGADT7-T, pGBKT7-lam, and pGBKT7-p53 were purchased from Clontech (Mountain View, CA, USA). Yeast strain AH109 was co-transformed with the combination of the pGADT7 and the pGBKT7 plasmids. Transformed yeast cells containing both plasmids were first grown on SD-minus Trp/Leu plates (DDO) to maintain the two plasmids and then were subcloned replica plated on SD-minus Trp/Leu/Ade/His plate (QDO).

**ASC oligomerization**. The supernatants of cell lysates were mixed with SDS-loading buffer for western blot analyses using an antibody against ASC. The pellets were washed with PBS three times and were cross-linked using fresh DSS (2 mM, Sigma) at 37 °C for 30 min. Then the cross-linked pellets were centrifuged and mixed with SDS-loading buffer for western blot analyses.

**Statistical analyses**. All experiments were reproducible and repeated at least three times with similar results. Samples were analyzed by one-way analysis of variance with Tukey's post-hoc test. Abnormal values were eliminated using a follow-up

Grubbs test. A Levene's test for equality of variances was performed, which provided information for Student's *t*-tests to distinguish the equality of means. Means were illustrated using histograms, with error bars representing standard error of the mean (s.e.m); values of $P < 0.05$ were considered to indicate statistical significance.

**Data Availability**. The authors declare that all data supporting the findings of this study are available within the paper and its supplementary information files or are available from the authors upon request.

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

## Acknowledgements

This work was supported by National Natural Science Foundation of China (No. 81730061, 31230005, 81471942, 31200134, and 31270206); the National Mega Project on Major Infectious Disease Prevention (No. 2017ZX10103005); and the Major Program of Guangdong Natural Science Foundation of China (No. S201230006598). We thank Dr. Cheng-Feng Qin of the Beijing Institute of Microbiology and Epidemiology, China for the gift of 129/Sv/Ev mice (A129 mice) deficient in IFN-α/β receptors; Dr. Di Wang of Zhejiang University School of Medicine, China for kindly providing C57BL/6 *NLRP3*⁻/⁻ mice; Dr Takaji Wakita of the National Institute of Infectious Diseases, Japan for the gift of hepatitis C virus genotype 2a strain JFH-1; and Dr. Pei-Yong Shi of the University of Texas Medical Branch, USA for the gift of the infectious cDNA clone of ZIKV.

## Author contributions

W.W., G.L., D.W., Z.L., K.W., X.L., F.L., Y.L., and J.W. contributed to the design of experiments. W.W., G.L., D.W., Z.L., P.P., M.T., Y.W., F.X., A.L., and L.R. contributed to the conduction of experiments. G.L., D.W., X.L., Z.L., A.L., and L.R. contributed to

procurement of reagents. W.W., G.L., X.L., F.L., Y.L., and J.W. contributed to writing the paper. W.W. and J.W. contributed to editing the paper.

## Additional information

**Competing interests:** The authors declare no competing financial interests.

