## [Peer Review File · Nature Communications]

Reviewers' comments:

Reviewer #1 (Remarks to the Author):

Wang and colleagues study NLRP3 inflammasome activation directly by ZIKV protein NS5. There is a significant amount of data to support the claims, and most control experiments are performed appropriately. The data add to the field of knowledge around ZIKV dependent inflammation and could be significant in understanding the pathogenesis of this virus.

Major points

- Usage of CHX to argue that NLRP3 inflammasome activation is due to inhibition of ZIKV protein synthesis may not be appropriate. How is it possible to argue that this is not an effect on the synthesis of a host protein? Although nigericin stimulation is unaltered, that pathway may be distinct from the one activated by ZIKV.
- The experiments in 4e-g are not compelling. Activating NLRP3 first with NS5, next with ATP, is convoluted. It is not clear why the baseline difference is then totally gone when poly(dA:dT) is used as well? Perhaps it is more appropriate just to focus on NS5 induced IL-1b, and treat with inhibitors such as MCC950? Also, isn't this spontaneous IL-1b from NS5 alone worrying? Previously the authors argued that priming was due to ZIKV RNA, but in this experiment where does the priming come from?
- Does NS5 form an amyloid or misfold as other potential triggers for NLRP3 (A-beta, IAPP) are known to do? The authors could stain with ThT or perform some other analysis for amyloid at the centre of these "spots"?
- All blots need molecular weight markers and the full blot should be included in supp data
- Cell imaging requires larger images with more cells, to be sure that the images selected are representative. Can be presented in supp data.

Minor points

- Page 4 line 71, IFN-1b instead of IL-1b
- NLRP3 "spot" could be instead "speck" to be consistent with ASC inflammasome literature?
- Figure 4c is mislabeled

Reviewer #2 (Remarks to the Author):

Title: Zika virus infection induces host inflammatory responses by facilitating NLRP3 inflammasome assembly and interleukin-1b secretion

The authors present their studies of Zika virus (ZIKV), providing evidence that ZIKV induced IL-1beta in vivo (identified in human specimen) and then work to examine ZIKV/NLRP3 interactions as the driver of IL-1beta induction. Of note is that the IL-1beta levels presented throughout the study are very small and likely insignificant for driving an actual inflammatory phenotype in vivo, and for this and other reasons listed below there are complications that need to be addressed to fully support the author's conclusions.

Note

This group used ZIKV isolate z16006 (GenBank accession number KU955589.1). According to GenBank, data for this isolate was collected 2/16/2016 and the genotype is described as "Asian"; BLAST query revealed 100% sequence similarity with other Chinese ZV strains from 2016 and 99% similarity with a Puerto Rico strain from 2015. One important item is whether or not the authors have tested for mycoplasma contamination of their ZIKV stocks. This is important question because mycoplasma is a major agonist of NLRPP3 and many labs in the field have found their ZIKV stocks heavily contaminated with mycoplasma (please note that observation is not published but is clearly dominating the concerns of the field)- the authors first and foremost need to show conclusively that their virus stocks and cells are not contaminated with mycoplasma. Please provide assay results to show that 1) the assay has both + and - controls, and 2) the assay is run on the ZIKV and cells used in these studies. Controlling for mycoplasma contamination is not directly addressed in the methods section. Virus used in infections was propagated in Vero or C6/36 cells.

-Was the virus filtered or sucrose purified following harvest?

-Were any QC measures used to ensure viral stocks were mycoplasma-free?

Figure 1: The effects of ZIKV infection on IL-1 β production and secretion:

a. Though low overall, the average IL-1b levels statistically significantly higher in sera of 11 ZV infected humans than 13 healthy controls

- samples were acquired from China's CDC

- Was day/stage of ZV infection documented?
- Were all infected persons symptomatic?
- Were viral strains sequenced or otherwise identified?
- How were patients identified as ZV positive?

b. Viral titers in blood of AG129 mice infected w/ ZV $\sim 10^7$ by 2 dpi, are down to 10^3 by 6 dpi- how are the mice controlling ZIKV replication despite lacking innate and T cell defenses? This data is highly inconsistent with the field which has shown that lacking innate defenses of interferon of both type 1 and 2 IFNs results in loss of virus restriction in mice and the ZIKV then grows to overwhelmingly high titers. Thus, these results as shown are highly controversial.

• Whole blood or serum? Says "blood" in figure legend, "Sera" in Figure- which is correct?

c. IL-1b levels increased out to 4 dpi in sera of ZV-infected AG129 mice, then subsided- this could be tracking with viral replication in the AG129 mice but a major problem here is that the mice are compromised and what inflammatory induction we see in these mice is actually artificial in a sense because the virus is normally highly restricted and without disease in WT mice, where IL-1beta has not been demonstrated. .

d-g. ZV infection induced very low/modest IL-1b mRNA transcription and IL-1b release (signals 1 and 2) in human PBMC's- thus the IL-1beta production ex vivo is not impressive. OF note is that deciduitis (inflammation of the uterus, where the major ZIKV is present in human pregnancy- has not been reported in the now 100's of ZIKV pregnancies that have been linked to clinical reports through the Americas. This lack of inflammatory disease is apparent and does not support the authors observations overall in a real life phenotype setting. The authors need to comment on this discrepancy.

-The authors Tested different MOI's, timepoints, used LPS (1 ug/mL) as positive control for signals 1 and 2 but no statistics are presented for the qRT PCR data.

h-k. ZIKV infection of THP-1 macs induced IL-1b mRNA transcription and IL-1b release; more IL-1b at MOI: 1 and 48 hpi is shown.

- They used nigericin as positive control for signal 2, but not signal 1- this is puzzling because nigericin is signal 1 agonist. Please explain .

l-m. Western blots of ZV-infected THP-1 macs

- Showed pro-IL-1beta levels increasing following ZV infection – good

Figure 2:

- Nigericin used as control

- Why is cleaved caspase-1 showing up in sup of shNLRP3 cells in Fig. 2d? This should not happen.

- Western blot showed rather poor levels of NLRP3 knockdown in Fig. 2f, though levels of released IL-1b were still markedly reduced. Please repeat this experiment using cells with full knockdown of NLRP3.

g-h. ZV-infected BMDC's and BMDMs from wt C57BL/6 mice released IL-1b, while those from NLRP3 ko mice did not.

- A problem here is the IL-1b release levels in ZIKV-infected BMDMs were very small (~15 pg/mL)- this is not even physiologically relevant.

- Judging from the treatment conditions and data, it appears that BMDCs and BMDMs were derived from non-treated mice, then infected with ZV. However, legends for both Figure 2 and Supp. Fig. 2 seem to state that mice were first treated or infected with ZV, then cells were differentiated. This should be clarified and/or changed.

m. ZV induced ASC speck formation in THP-1 macs.

Why are there multiple ASC specks in cell shown in Fig. 2m? (should generally be 1 per cell)

Only 1-3 cells shown per field of view in Fig. 2j-m. Data would be more compelling if cells with NLRP3 or SC specks were quantified.

- Also, how do researchers know that ZIKV infection directly induces NLRP3 or ASC speck formation in these cells? It would be good to also stain for ZIKV to confirm that cells containing ASC specks are also infected with ZIKV.

Figure 3:

g. UV or heat inactivated ZV did not induce release of cleaved IL-1b or caspase-1, as seen by WB, while live ZV did

- Which cells are these: PBMCs or THP-1s?

h. ZV protein production is required for IL-1b maturation and release

- Cycloheximide pretreatment abrogate IL-1b release induced in ZV-infected but not nigericin-treated THP-1 macs.

- CHX would also prevent production of pro-IL-1b induced by ZV infection, thereby also reducing the amount of processed IL-1b that is released; please assess and show pro-IL-1b levels by Westn blot.

i. Transfection of THp-1 macs with ZIKV genomic RNA induced IL-1b mRNA transcription (~3 fold), but not IL-1b release.

- How long after transfection were these samples harvested?

Transfected ZIKV viral RNA should be infectious and initiate productive infection- did this occur? Please show evidence of productive infection.

- Please show Western blot for viral proteins.
- If viral proteins are expressed in transfected cells, then why is there no IL-1b release?

Figure 4: ZIKV NS5 facilitates the activation of NLRP3 inflammasome

-Does NS5 over ride the need for a signal 1 and signal 2 in NLRP3 inflammasome activation?
Please explain and show data to support this notion.

- What is the purpose of the poly(dA;dT) control?

Figure 5: NS5 binds to NLRP3 by interacting with the NACHT and LRR domains of NLRP3

-In 293T cells, NS5 immuno-precipitated with NLRP3, but not ASC or Caspase-1; but in Fig. 5b, is IgG light chain overlapping with the same band as caspase-1? (If so, would be good to label the light chain band).

In 293T and Hela cells, NS5 translocated from nucleus and co-localized with NLRP3, NACHT, and LRR domains to form spots in cytosol. (HeLa and NLRP3 truncation mutant data is in supplementary Fig. 6)

However, NS5 did not co-localize with ASC, Casp1, or NLRP3 pyrin domain- why is this?

-The observation of ring-like structures, with NS5 on inside, NLRP3 in middle, ASC on outside could this be an artifact of transfection/overexpression- these structure should be observed in primary cells, such as the cultured monocyte derived macrophages without overexpression. Note that overexpression of inflammasome components will typically induce such structures to form even without stimulus. Thus these data are highly questionable. Do these structures also show up in THP1 cells nontransfected but simply infected with ZIKV?

- Were NS5-NLRP3-ASC co-localization and ring-like structures observed in THP-1 macrophages stably expressing NS5?

Was ASC speck formation also observed by IFA in infected THP-1 macrophages?

Figure 7: ZIKV infection induces inflammatory responses in mice

-A129 mice lacking IFN a/b receptors were susceptible to ZIKV infection

-Neutrophil/mononuclear cell infiltration in brain

-Corpuscle structures in spleen were large and poorly defined

-Increased vascular permeability in livers and kidneys

-Weight loss began 4 dpi

-All mice died 9-12 dpi

-Authors concluded that ZIKV induced severe inflammatory responses in mice but note the immunocompromised nature of the A129 mice and one cannot make such strong conclusions without much more analysis. Are these inflammatory responses dependent on IL-1b or NLRP3? The authors need to place effort on generating knockout NLRP3 or IL-1b in A129 mice via breeding with existing NLRP3 and IL-1r KO lines to properly generate and evaluate inflammatory phenotype in response to ZIKV infection. Otherwise, the observed inflammatory phenotype may simply be driven by cell death or release of other cytokines/DAMPs.

It is understood what the researchers' desired to use A129 mice, as wt C57BL/6 mice with robust IFN responses generally do not succumb to ZIKV infection, but the presented data does not convincingly demonstrate that inflammation in response to in vivo ZV infection is NLRP3 or inflammasome-dependent.

Thus, overall the study has some major areas of concern and below are suggested places to focus key revisions:

Repeat experiments showing virus-induced IL-1 β maturation and release in THP-1 macs, PBMCs, or BMDCs using sucrose purified ZIKV that is demonstrated to be mycoplasma-free.

Show basic virologic analysis of infected PBMCs, BMDCs, or THP-1 macs, including percent cell infectivity and plaque assays showing whether or not infectious viral particles are generated in these cells. Showing viral RNA levels alone is not sufficient.

Show that the NLRP3 inflammasome is necessary for the inflammatory phenotype observed in A129 mice. Include data from wt and NLRP3 ko C57BL/6 mice.

As well as some other important revisions mentioned in the review above. (very importantly, include new IFA data showing cells staining positive for ZIKV protein and ASC specks)

Point-by-Point Response to the Referees' Comments

Reviewer #1 (Remarks to the Author):

Wang and colleagues study NLRP3 inflammasome activation directly by ZIKV protein NS5. There is a significant amount of data to support the claims, and most control experiments are performed appropriately. The data add to the field of knowledge around ZIKV dependent inflammation and could be significant in understanding the pathogenesis of this virus.

Authors' Response: Thank you very much for your remarks.

Reviewer #1's Major points

- Usage of CHX to argue that NLRP3 inflammasome activation is due to inhibition of ZIKV protein synthesis may not be appropriate. How is it possible to argue that this is not an effect on the synthesis of a host protein? Although nigericin stimulation is unaltered, that pathway may be distinct from the one activated by ZIKV.

Authors' Responses: Thank you for the comments.

Our result shows that IL-1 β is produced normally in response to the stimulation of nigericin in the cells pretreated with cycloheximide (CHX) (a protein synthesis inhibitor), indicating that NLRP3 inflammasome activation does not require de novo protein translation. This result is consistent with previous report, which demonstrated that acute NLRP3 inflammasome activation is independent of new protein synthesis (Lin et al., PNAS USA 2014; 111(2):775-780. PMID:24379360.). We also show that in the presence of CHX, IL-1 β secretion, IL-1 β cleavage, and Casp-1 cleavage are activated by Nigericin but not by ZIKV, suggesting that production of ZIKV encoded proteins is required for IL-1 β activation. This result is consistent with a previous study, which revealed that pretreatment of cells with CHX significantly inhibits EMCV-induced IL-1 β secretion and indicated that virus-encoded proteins are required for the activation of the NLRP3 inflammasome (Ito et al., PLoS Pathog 2012; 8(8):e1002857. PMID:22916014.).

According to your suggestion, we have carried out additional experiments to

determine the effect of CHX on the production of pro-IL-1 β and pro-Casp-1. The results indicate that the production of pro-IL-1 β and pro-Casp-1 is attenuated by CHX in the absence of Nigericin (Revised Fig. 3h, lower panel, lane 3 vs. 2) and the presence of Nigericin (Revised Fig. 3h, lower panel, lane 6 vs. 5). However, the secretion and cleavage of IL-1 β is not affected by CHX in the presence of Nigericin (Revised Fig. 3h, upper panel, lane 6 vs. 5). More interestingly, the secretion and cleavage of IL-1 β is attenuated by CHX upon ZIKV infection (Revised Fig. 3h, upper panel, lane 9 vs. 8). These results are agreed with your suggestion that nigericin stimulation is distinct from the one activated by ZIKV. Taken together, we suggest that de novo protein synthesis is not required for Nigericin-induced activation of IL-1 β , but required for ZIKV-induced activation of IL-1 β .

- The experiments in 4e-g are not compelling. Activating NLRP3 first with NS5, next with ATP, is convoluted. It is not clear why the baseline difference is then totally gone when poly(dA:dT) is used as well? Perhaps it is more appropriate just to focus on NS5 induced IL-1b, and treat with inhibitors such as MCC950? Also, isn't this spontaneous IL-1b from NS5 alone worrying? Previously the authors argued that priming was due to ZIKV RNA, but in this experiment where does the priming come from?

Authors' Responses: Thank you for the comments.

The results from Fig. 4e-g suggest that NS5 facilitates ATP in the activation of NLRP3, but does not facilitate poly(dA:dT) in the activation of AIM2, since ATP is specific for NLRP3 inflammasome, whereas poly(dA:dT) is specific for AIM2 inflammasome. Our results indicate that NS5 is specific for NLRP3 inflammasome. We agree with you that it is more appropriate just to focus on NS5 induced IL-1 β , and thus we have removed the results of poly(dA:dT) from the revised manuscript.

According to your suggestion, we have carried out additional experiments to determine the role of NS5 in the induction of IL-1 β . The cells stably expressing NS5 are treated with a Caspase-1 inhibitor (Ac-YVAD-cmk). The results show that IL-1 β secretion (Revised Fig. 4h) as well as IL-1 β and Casp-1 cleavages (Revised Fig. 4i)

are induced by NS5, but NS5-induced activations of IL-1 β and Casp-1 are inhibited by Ac-YVAD-cmk, further demonstrating that NS5 facilitates the activation of NLRP3 inflammasome.

Activation of NLRP3 inflammasome is regulated by two pathways. One is regulated by the activation of NF- κ B pathway, which subsequently induces the expression of pro-IL-1 β , NLRP3, and ASC. The other is initiated by the assembly of NLRP3 inflammasome, which results in the activation of Casp-1 and the maturation of IL-1 β .

We have demonstrated that ZIKV genomic RNA induces the NF- κ B pathway and suggested that the priming is due to ZIKV RNA (Supplementary Fig. 3e and f). However, the experiments in Fig. 4e-g are designed to determine the role of NS5 in the assembly and activation of NLRP3 inflammasome, but not the activation of NF- κ B pathway or priming. In addition, we carried out an additional experiment to determine whether NS5 plays a role in the priming of IL-1 β . The result shows that NS5 facilitates the secretion of IL-1 β , but not the expression of pro-IL-1 β (Revised Fig. 4f).

- Does NS5 form an amyloid or misfold as other potential triggers for NLRP3 (A-beta, IAPP) are known to do? The authors could stain with ThT or perform some other analysis for amyloid at the centre of these “spots”?

Authors’ Responses: Your suggestion is very good. However, it has not been reported that NS5 can form an amyloid. In this study, we focus on the function of NS5, but not the structure of NS5, in the activation of NLRP3. It would be interesting to determine whether NS5 is misfold during the activation of NLRP3 in the future.

- All blots need molecular weight markers and the full blot should be included in supp data

Authors’ Response: According to your suggestion, we have added the molecular weight markers to all blots and included the full blot in the supplementary data.

- Cell imaging requires larger images with more cells, to be sure that the images selected are representative. Can be presented in supp data.

Authors' Response: Thank you for the suggestion. We have provided larger images for the cell imaging with more cells in the supplementary data.

Reviewer #1's Minor points

- Page 4 line 71, IFN-1b instead of IL-1b

Authors' Response: Thanks. We have changed “IFN-1 β ” to “IL-1 β ” in the revised manuscript.

- NLRP3 “spot” could be instead “speck” to be consistent with ASC inflammasome literature?

Authors' Response: Thank you. We have changed the “spot” to “speck” in the revised manuscript.

- Figure 4c is mislabeled

Authors' Response: Thank you again. We have corrected it in the revised manuscript.

Reviewer #2 (Remarks to the Author):

The authors present their studies of Zika virus (ZIKV), providing evidence that ZIKV induced IL-1beta in vivo (identified in human specimen) and then work to examine ZIKV/NLRP3 interactions as the driver of IL-1beta induction. Of note is that the IL-1beta levels presented throughout the study are very small and likely insignificant for driving an actual inflammatory phenotype in vivo, and for this and other reasons listed below there are complications that need to be addressed to fully support the author's conclusions.

Authors' Responses: Thank you for the remarks.

In order to answer your question, we have repeated the experiments in human PBMCs. The results show that ZIKV infection significantly induces IL-1 β secretion

by 70-fold and IL-1 β mRNA expression by 30-fold (Revised Fig. 1d–g). We have also calculated the results in differentiated THP-1 cells, in which ZIKV infection induces IL-1 β secretion by 10-fold and IL-1 β mRNA expression by 10-fold (Revised Fig. 1h–k). In addition, IL-1 β secretion is not detected in the supernatant of mice BMDC and BMDM, but IL-1 β secretion is detected at the concentration of 200 pg/ml in the supernatant of ZIKV-infected mice BMDC and mice BMDM (Revised Fig. 2g and i), indicating that ZIKV infection significantly induces IL-1 β secretion. Taken together, all of our results demonstrate that ZIKV infection significantly induces the secretion of IL-1 β in infected human specimen, culture cells, and mice.

Reviewer #2's Note

This group used ZIKV isolate z16006 (GenBank accession number KU955589.1). According to GenBank, data for this isolate was collected 2/16/2016 and the genotype is described as “Asian”; BLAST query revealed 100% sequence similarity with other Chinese ZV strains from 2016 and 99% similarity with a Puerto Rico strain from 2015. One important item is whether or not the authors have tested for mycoplasma contamination of their ZIKV stocks. This is important question because mycoplasma is a major agonist of NLRPP3 and many labs in the field have found their ZIKV stocks heavily contaminated with mycoplasma (please note that observation is not published but is clearly dominating the concerns of the field)- the authors first and foremost need to show conclusively that their virus stocks and cells are not contaminated with mycoplasma. Please provide assay results to show that 1) the assay has both + and - controls, and 2) the assay is run on the ZIKV and cells used in these studies.

Controlling for mycoplasma contamination is not directly addressed in the methods section. Virus used in infections was propagated in Vero or C6/36 cells.

Authors' Response: Thank you for the note.

-Was the virus filtered or sucrose purified following harvest?

Authors' Response: Yes, the virus was filtered by 0.45 μ m filter for cell infection,

and purified by 26% sucrose following harvest for mice infection. We added the experimental procedures to the revised Materials and Method section.

-Were any QC measures used to ensure viral stocks were mycoplasma-free?

Authors' Responses: We have tested for mycoplasma contamination of the stocks of ZIKV and cells used in this study. The PCR result, as shown in the following figure, demonstrates that the stocks of ZIKV and cells (Vero and C6/36) are free of mycoplasma by using MycoTest kit which is purchased from ChanGEnome for the detecting mycoplasma.

In addition, transmission electron microscope (TEM) result, as shown in the following figure, also demonstrates that the stock of ZIKV is free of mycoplasma.

Reviewer #2's comments: Figure 1: The effects of ZIKV infection on IL-1 β production and secretion:

1a. Though low overall, the average IL-1b levels statistically significantly higher in sera of 11 ZV infected humans than 13 healthy controls.

Authors' Response: Good point!

- samples were acquired from China's CDC

Authors' Response: Samples are acquired from Guangdong's CDC.

• Was day/stage of ZV infection documented?

Authors' Response: Yes, day/stage of ZV infection were documented (Please see the following table).

patient number	Gender	age	date of onset	sample collection date	RT-PCR detect		
					Blood	Urine	Saliva
GDZ16001	male	28	2016.02.10	2016.02.12	-	+	+
GDZ16006	male	38	2016.02.14	2016.02.16	+	-	-
GDZ16021	male	6	2016.02.24	2016.02.26	+	+	+
GDZ16022	female	8	2016.02.26	2016.02.26	-	+	+
GDZ16195	male	47	2016.03.01	2016.03.05	+	+	+
GDZ16264	male	19	2016.03.07	2016.03.09	-	+	+
GDZ16460	female	13	2016.03.23	2016.03.28	no sample	+	+
GDZ16461	female	12	2016.03.29	2016.03.29	+	-	+
GDZ16473	female	7	2016.04.04	2016.04.06	-	+	+
GDZ16482	male	37	2016.05.08	2016.05.13	+	-	-
GDZ16487	female	40	2016.07.07	2016.07.09	-	+	+

• Were all infected persons symptomatic?

Authors' Response: Yes, all infected persons were symptomatic. The information of 11 ZIKV infected patients has been previously published online (DOI:10.3760/cma.j.issn.0254-5101.2016.10.001).

• Were viral strains sequenced or otherwise identified?

Authors' Response: The information has been described in the Materials and Methods section of the previous version [The ZIKV isolate z16006 (GenBank accession number, KU955589.1) isolated by the Institute of Pathogenic Microbiology, Center for Disease Control and Prevention of Guangdong (Guangzhou, Guangdong, China), was used in this study].

• How were patients identified as ZV positive?

Authors' Response: These patients were identified as ZIKV positive by ZIKV RT-PCR detection reagent kit (purchased from daan gene), which has been published

previously online (DOI:10.3760/cma.j.issn.0254-5101.2016.10.001).

1b. Viral titers in blood of AG129 mice infected w/ ZV $\sim 10^7$ by 2 dpi, are down to 10^3 by 6 dpi- how are the mice controlling ZIKV replication despite lacking innate and T cell defenses? This data is highly inconsistent with the field which has shown that lacking innate defenses of interferon of both type 1 and 2 IFNs results in loss of virus restriction in mice and the ZIKV then grows to overwhelmingly high titers. Thus, these results as shown are highly controversial.

Authors' Responses: Thank you for the comment.

It is an important question how the mice control ZIKV replication despite lacking innate immunity. We reveal that ZIKV replication is eliminated or inhibited in the blood of A129 mice lacking type I IFNs (Revised Fig. 1b), and show that the sera levels of IL-1 β are increased rapidly until 4 days post-infection and decreased thereafter (Revised Fig. 1c), suggesting that ZIKV induces IL-1 β production and secretion and indicating that IL-1 β may attenuate ZIKV replication in the blood of A129 mice. This result is consistent with a previous report demonstrating that ZIKV titer in the serum is observed to peak on day two and can't be detected on day six in all AG129 mice infected with ZIKV (Aliota et al., PLoS Negl Trop Dis 2016;10(4):e0004682. PMID:27093158).

In order to further confirm this result, we carried out additional experiments to determine the effect of IL-1 β on ZIKV replication in the mice. The results show that At 6 days post-infection and in the blood of infected mice, the level of ZIKV titer (3.7×10^3) was much lower in the absence of Ac-YVAD-cmk (Revised Fig. 8a) as compared to that (4×10^5) in the presence of Ac-YVAD-cmk (Revised Fig. 8b). These results indicate that inhibition of IL-1 β activation results in the enhancement of ZIKV replication and suggest that IL-1 β plays an important role in the attenuation of ZIKV replication.

Moreover, ZIKV replication (as indicated by ZIKV E protein production) is detected in the brain, spleen, liver, and kidney of ZIKV-infected mice (Revised Fig. 8c-f). In addition, ZIKV E protein in the brain of infected mice was reduced by

Ac-YVAD-cmk (Revised Fig. 8c), but ZIKV E protein in the spleen, liver, or kidney of infected mice was enhanced by Ac-YVAD-cmk (Revised Fig. 8d–f), demonstrating that Casp-1 plays an important role in the regulation of ZIKV replication in mice and that ZIKV replication is regulated by caspase-1 or IL-1 β . Taken together, we demonstrate that IL-1 β plays an important role in the inhibition of ZIKV replication.

•Whole blood or serum? Says “blood” in figure legend, “Sera” in Figure- which is correct?

Authors’ Response: Thank you for the comment. We have changed the word “sera” to “blood” in the in Figures of revised manuscript.

1c. IL-1 β levels increased out to 4 dpi in sera of ZV-infected AG129 mice, then subsided- this could be tracking with viral replication in the AG129 mice but a major problem here is that the mice are compromised and what inflammatory induction we see in these mice is actually artificial in a sense because the virus is normally highly restricted and without disease in WT mice, where IL-1 β has not been demonstrated

Authors’ Responses: Thank you for the comment. It is unfortunately that ZIKV cannot infect WT mice, where IL-1 β has not been demonstrated. Fortunately, A129 mice is a well-established animal model for the study of the infection of viruses, including ZIKV. Several research groups have used this mice model to study ZIKV infection and pathogenesis. For examples, ZIKV infection leads to the development of microcephaly in A129 mice (Li et al., *Cell Stem Cell* 2016;19(1):120-126. doi: 10.1016/j.stem.2016.04.017.), and ZIKV causes testis damage and leads to male infertility in A129 mice (Ma et al., *Cell* 2016;167(6):1511-1524.e10. doi: 10.1016/j.cell.2016.11.016.). In this study, we demonstrate that IL-1 β levels are not only significantly higher in the sera of ZIKV infected humans (Revised Fig. 1a), but also increased rapidly in the sera of ZIKV infected A129 mice (Revised Fig. 1c). These results suggest that IL-1 β may play an inhibitory role in the blood of both human and mice even in the absence of type I IFNs. We also show that inhibition of IL-1 β activation by Ac-YVAD-cmk results in the enhancement of ZIKV replication

(Revised Fig. 8a and b), suggesting that IL-1 β plays an important role in the attenuation of ZIKV replication.

1d-g. ZV infection induced very low/modest IL-1 β mRNA transcription and IL-1 β release (signals 1 and 2) in human PBMC's- thus the IL-1 β production ex vivo is not impressive. Of note is that deciduitis (inflammation of the uterus, where the major ZIKV is present in human pregnancy- has not been reported in the now 100's of ZIKV pregnancies that have been linked to clinical reports through the Americas. This lack of inflammatory disease is apparent and does not support the authors observations overall in a real life phenotype setting. The authors need to comment on this discrepancy.

Authors' Responses: Thank you for the comments and note.

In order to answer your question, we have repeated the experiments in human PBMCs. The results show that ZIKV infection significantly induces IL-1 β secretion by 70-fold and IL-1 β mRNA expression by 30-fold (Revised Fig. 1d–g). We have also calculated the results in differentiated THP-1 cells, in which ZIKV infection induces IL-1 β secretion by 10-fold and IL-1 β mRNA expression by 10-fold (Revised Fig. 1h–k). In addition, IL-1 β secretion is not detected in the supernatant of mice BMDC and BMDM, but IL-1 β secretion is detected at the concentration of 200 pg/ml in the supernatant of ZIKV-infected mice BMDC and mice BMDM (Revised Fig. 2g and i), indicating that ZIKV infection significantly induces IL-1 β secretion. Taken together, all of our results demonstrate that ZIKV infection significantly induces the secretion of IL-1 β in infected human specimen, culture cells, and mice.

We agree with your note that deciduitis (inflammation of the uterus) has not been reported in ZIKV infected pregnancies. It is possible that the pathogenesis caused by ZIKV infection in pregnancies is not due to the inflammatory disease, but may be due to direct viral infection of the fetuses. In this study, we do not explore the correlation between deciduitis and ZIKV infection during human pregnancy. However, there are several reports have revealed the correlation between IL-1 β and inflammatory disease. For examples, IL-1 β induces the production of prostaglandin

E2 in the hypothalamus that resets the hypothalamic thermostat to induce fever (Dinarello et al., PNAS USA 1977; 74(10):4624-4637. PMID:22079.), IL-1 β is involved in the afferent pain response by signaling inflammatory pain to the hypothalamus (Ferreira et al., Nature 1988; 334(6184):698-700. PMID:3137474), and overproduction of IL-1 β induces a number of genetically determined auto-inflammatory disorders (Stojanov and Kastner, Curr Opin Rheumatol 2005; 17(5):586-599. PMID:16093838). These diseases usually manifest as fever, high acute-phase responses, arthritis, and rash, the symptoms similar to those of ZIKV-associated illness.

-The authors Tested different MOI's, timepoints, used LPS (1 ug/mL) as positive control for signals 1 and 2 but no statistics are presented for the qRT PCR data.

Authors' Response: The qRT-PCR is performed by using the Roche LC40 and SYBR RT-PCR kits. We perform three repeats for each sample. The result shows the mean with SEM that is used to create the graph. Although this method can't be used to calculate the statistics for the qRT-PCR data, it has been used in many publications (Teo et al., Nat Cell Biol 2010; 12(8):758-567. PMID:20622870; Gurcel et al., Cell. 2006; 126(6):1135-1145. PMID:16990137; Ghosh et al., Nat Cell Biol 2012; 14(12):1270-81. PMID:23159929).

1h-k. ZIKV infection of THP-1 macs induced IL-1b mRNA transcription and IL-1b release; more IL-1b at MOI: 1 and 48 hpi is shown.

- They used nigericin as positive control for signal 2, but not signal 1- this is puzzling because nigericin is signal 1 agonist. Please explain .

Authors' Responses: Thank you for the comment. Nigericin is a microbial toxin derived from *Streptomyces hygroscopicus* and acts as a potassium ionophore. The release of IL-1 β in response to nigericin has been demonstrated to be NLRP3-dependent (Mariathasan et al., Nature 2006; 440(7081):228-32. PMID:16407890.). Similar to ATP, nigericin induces a net decrease in intracellular levels of potassium, which is crucial for the activation of caspase-1. Nigericin requires

signaling through pannexin-1 to induce caspase-1 maturation and IL-1 β processing and release (Pelegrin and Surprenant. J Biol Chem 2007; 282(4):2386-2394. PMID: 17121814.). Nigericin alone can't induce IL-1 β release in BMDM. The TLR-primed pathway is necessary for Nigericin (Mariathasan et al., Nature 2006; 440(7081):228-232. PMID:16407890.). Our result also demonstrates that Nigericin treatment can't induce the expression of IL-1 β mRNA and protein, and thus Nigericin acts as a positive control for signal 2, but not signal 1.

11-m. Western blots of ZV-infected THP-1 macs

- Showed pro-IL-1beta levels increasing following ZV infection – good

Authors' Responses: Thank you for the nice comment.

Reviewer #2's comments: Figure 2:

- Nigericin used as control

• Why is cleaved caspase-1 showing up in sup of shNLRP3 cells in Fig. 2d? This should not happen.

Authors' Responses: Sorry for the confusion. The cleaved caspase-1, which is showing up in sup of shNLRP3 cells in Fig. 2d, is nonspecific (Please see the following original picture). We have repeated this experiment to get a more clear result, which shows that cleaved caspase-1 is not detected (Revised Fig. 2d).

• Western blot showed rather poor levels of NLRP3 knockdown in Fig. 2f, though levels of released IL-1b were still markedly reduced. Please repeat this experiment using cells with full knockdown of NLRP3.

• Actually, we did this experiment using the same cells for the treatment with Nigericin and ZIKV. The cells are full knockdown of NLRP3 by sh-RNA. A lower

exposure of the Western blot result now shows that the level of NLRP3 is much lower in NLRP3-knockdown cells in revised Fig. 2f of the revised manuscript.

2g-h. ZV-infected BMDC's and BMDMs from wt C57BL/6 mice released IL-1b, while those from NLRP3 ko mice did not.

- A problem here is the IL-1b release levels in ZIKV-infected BMDMs were very small (~15 pg/mL)- this is not even physiologically relevant.

Authors' Response: Thank you for the comment. We agree with you that a problem here is the IL-1 β release levels in ZIKV-infected BMDMs were very small (~15 pg/mL). We have repeated this experiment more carefully. The new results shown in the revised Fig. 2i indicates that the IL-1 β release levels in ZIKV-infected BMDMs is now much higher (~110 pg/mL).

•Judging from the treatment conditions and data, it appears that BMDCs and BMDMs were derived from non-treated mice, then infected with ZV. However, legends for both Figure 2 and Supp. Fig. 2 seem to state that mice were first treated or infected with ZV, then cells were differentiated. This should be clarified and/or changed.

Authors' Responses: Thanks for the comments. We have described the treatment conditions and data in the Materials and Methods section. Mice BMDCs were differentiated from fresh bone marrow cells of A129 mice, C57BL/6 WT mice, and C57BL/6 NLRP3^{-/-} mice in RPMI 1640 medium containing 10% heat-inactivated fetal bovine serum (FBS) in the presence of granulocyte macrophage colony-stimulating factor (GM-CSF) in six-well plates for 5 days. The culture medium was replaced every other day. Mouse BMDMs were differentiated from fresh bone marrow cells of A129 mice, C57BL/6 WT mice, and NLRP3^{-/-} mice. The bone marrow cells were incubated in six-well plates for 10 days with 10% L929-conditioned, 10% heat-inactivated FBS in RPMI 1640 medium. The culture medium was replaced every 2 days. After treatment with GM-CSF or L929 supernatant for several days, the cells were infected with ZIKV. We have been clarified this in the revised legends for both Figure 2 and Supp. Figure S2.

•Why are there multiple ASC specks in cell shown in Fig. 2m? (should generally be 1 per cell)

Authors' Responses: Our results are consistent with previous studies that have demonstrated that NLRP3 and ASC can form multiple specks in THP-1 cells (Fernandes-Alnemri et al., Nature 2009; 458(7237):509-513. PMID:19158676; Bürckstümmer et al., Nat Immunol 2009; 10(3):266-272. PMID:19158679; Subramanian et al., Cell 2013; 153(2):348-361. PMID:23582325). The cell imaging that requires larger images with more cells is presented in supplementary data.

•Also, how do researchers know that ZIKV infection directly induces NLRP3 or ASC speck formation in these cells? It would be good to also stain for ZIKV to confirm that cells containing ASC specks are also infected with ZIKV.

Authors' Responses: We agree with you that it would be better to stain for ZIKV to confirm that cells containing ASC specks are also infected with ZIKV. However, the antibodies for ZIKV E, NLRP3, and ASC proteins used for Immunofluorescence assays are mouse monoclonal antibodies. Therefore, it is technically impossible for us to stain for ZIKV E, NLRP3, and ASC proteins in current conditions and in human cells. However, more importantly is that we have demonstrated in many experiments that ZIKV NS5 alone can induce the NLRP3 activation. Our results support that ZIKV infection can induce the activation of NLRP3 inflammasome and the speck formation of NLRP3 and ASC in these cells.

Reviewer #2's comments: Figure 3:

3g. UV or heat inactivated ZV did not induce release of cleaved IL-1b or caspase-1, as seen by WB, while live ZV did.

•Which cells are these: PBMCs or THP-1s?

Authors' Response: Thank you for the comment. Actually, we have mentioned in the figure legends that THP-1-differentiated macrophages are used in Fig. 3d–g.

-Cycloheximide pretreatment abrogate IL-1b release induced in ZV-infected but not

nigericin-treated THP-1 macs.

•CHX would also prevent production of pro-IL-1 β induced by ZV infection, thereby also reducing the amount of processed IL-1 β that is released; please assess and show pro-IL-1 β levels by Western blot.

Authors' Responses: Thank you for the comments.

Our result shows that IL-1 β is produced normally in response to the stimulation of nigericin in the cells pretreated with cycloheximide (CHX) (a protein synthesis inhibitor), indicating that NLRP3 inflammasome activation does not require de novo protein translation. This result is consistent with previous report, which demonstrated that acute NLRP3 inflammasome activation is independent of new protein synthesis (Lin et al., PNAS USA 2014; 111(2):775-780. PMID:24379360.). We also show that in the presence of CHX, IL-1 β secretion, IL-1 β cleavage, and Casp-1 cleavage are activated by Nigericin but not by ZIKV, suggesting that production of ZIKV encoded proteins is required for IL-1 β activation. This result is consistent with a previous study, which revealed that pretreatment of cells with CHX significantly inhibits EMCV-induced IL-1 β secretion and indicated that virus-encoded proteins are required for the activation of the NLRP3 inflammasome (Ito et al., PLoS Pathog 2012; 8(8):e1002857. PMID:22916014.).

According to your suggestion, we have carried out additional experiments to determine the effect of CHX on the production of pro-IL-1 β and pro-Casp-1. The results indicate that the production of pro-IL-1 β and pro-Casp-1 is attenuated by CHX in the absence of Nigericin (Revised Fig. 3h, lower panel, lane 3 vs. 2) and the presence of Nigericin (Revised Fig. 3h, lower panel, lane 6 vs. 5). However, the secretion and cleavage of IL-1 β is not affected by CHX in the presence of Nigericin (Revised Fig. 3h, upper panel, lane 6 vs. 5). More interestingly, the secretion and cleavage of IL-1 β is attenuated by CHX upon ZIKV infection (Revised Fig. 3h, upper panel, lane 9 vs. 8). These results are agreed with your suggestion that nigericin stimulation is distinct from the one activated by ZIKV. Taken together, we suggest that de novo protein synthesis is not required for Nigericin-induced activation of IL-1 β , but required for ZIKV-induced activation of IL-1 β .

3i. Transfection of THP-1 macs with ZIKV genomic RNA induced IL-1b mRNA transcription (~3 fold), but not IL-1b release.

- How long after transfection were these samples harvested?

Transfected ZIKV viral RNA should be infectious and initiate productive infection- did this occur? Please show evidence of productive infection.

Authors' Responses: Thank you for the comment. We have mentioned in the manuscript that THP-1-differentiated macrophages were treated with Lipo, stimulated with Lipo plus poly(dA:dT), or treated with Lipo and transfected with 1 or 5 µg/mL ZIKV genomic RNA for 6 h. The time (6 h) is too short for the production of the infectious ZIKV.

We have repeated this experiment in the revised manuscript. The result from plaque assay indicates that infectious ZIKV was not detected in the supernatants of treated THP-1-differentiated macrophages.

These results suggest that ZIKV genomic RNA itself can induce IL-1β mRNA expression, but not IL-1β activation and secretion.

- Please show Western blot for viral proteins.
- If viral proteins are expressed in transfected cells, then why is there no IL-1b release?

Authors' Responses: THP-1-differentiated macrophages were transfected with 1 or 5 µg/mL ZIKV genomic RNA for 6 h. The time (6 h) is too short for the production of the infectious ZIKV and for the expression of viral proteins. These results suggest that ZIKV genomic RNA itself, but not viral proteins, can induce IL-1β mRNA expression.

Figure 4: ZIKV NS5 facilitates the activation of NLRP3 inflammasome

-Does NS5 over ride the need for a signal 1 and signal 2 in NLRP3 inflammasome activation? Please explain and show data to support this notion.

Authors' Responses: Thank you for the comments.

Activation of NLRP3 inflammasome is regulated by two pathways. One is regulated by the activation of NF- κ B pathway, which subsequently induces the expression of pro-IL-1 β , NLRP3, and ASC. The other is initiated by the assembly of NLRP3 inflammasome, which results in the activation of Casp-1 and the maturation of IL-1 β .

We have demonstrated that ZIKV genomic RNA induces the NF- κ B pathway and suggested that the priming is due to ZIKV RNA (Supplementary Fig. 3e and f). However, the experiments in Fig. 4e-g are designed to determine the role of NS5 in the assembly and activation of NLRP3 inflammasome, but not the activation of NF- κ B pathway or priming. In addition, we carried out an additional experiment to determine whether NS5 plays a role in the priming of IL-1 β . The result shows that NS5 facilitates the secretion of IL-1 β , but not the expression of pro-IL-1 β (Revised Fig. 4f). Taken together, we suggest that ZIKV NS5 facilitates the activation of NLRP3 inflammasome through signal 2, but not signal 1.

•What is the purpose of the poly(dA:dT) control?

Authors' Response: Thank you for the comment. ATP is specific to NLRP3 inflammasome, whereas poly(dA:dT) is specific to AIM2 inflammasome. These results suggest that NS5 is specific to NLRP3 inflammasome. According to the Reviewer 1's comment, we have deleted the results of poly(dA:dT) in the revised manuscript.

Figure 5: NS5 binds to NLRP3 by interacting with the NACHT and LRR domains of NLRP3

-In 293T cells, NS5 immuno-precipitated with NLRP3, but not ASC or Caspase-1; but

in Fig. 5b, is IgG light chain overlapping with the same band as caspase-1? (If so, would be good to label the light chain band)

Authors' Responses: Yes, you are right. Caspase-1 is little smaller than IgG heavy chain. According to your suggestion, we have labeled the IgG heavy chain band in the Revised Fig. 5b.

In 293T and HeLa cells, NS5 translocated from nucleus and co-localized with NLRP3, NACHT, and LRR domains to form spots in cytosol. (HeLa and NLRP3 truncation mutant data is in supplementary Fig. 6)

However, NS5 did not co-localize with ASC, Casp1, or NLRP3 pyrin domain- why is this?

Authors' Responses: Thank you for the comment. Co-IP results demonstrate that NS5 can interact with NLRP3 protein, NLRP3 NACHT domain, and NLRP3 LRR domain, but fails to interact with ASC protein, Casp-1 protein, and NLRP3 PYRIN domain. Therefore, it is reasonable that in 293T and HeLa cells, NS5 is translocated from nucleus and co-localized with NLRP3, NACHT domain, and LRR domain to form spots in cytosol, but fails to be co-localized with ASC, Casp-1, or PYRIN domain. The results of immunofluorescence assays are consistent with the results of Co-IP assays.

Figure 6: The observation of ring-like structures, with NS5 on inside, NLRP3 in middle, ASC on outside could this be an artifact of transfection/overexpression- these structure should be observed in primary cells, such as the cultured monocyte derived macrophages without overexpression. Note that overexpression of inflammasome components will typically induce such structures to form even without stimulus. Thus these data are highly questionable. Do these structures also show up in THP1 cells nontransfected but simply infected with ZIKV?

Authors' Responses: Thank you for the comments.

The localization of NLRP3 as speck is an indication of inflammasome complex formation. As you suggested, we have investigated the effect of ZIKV on NLRP3

inflammasome formation in THP-1-differentiated macrophages (Revised Fig. 2j), HeLa cells (Revised Fig. 2k), and Vero cells (Revised Fig. 2l). In these cells, NLRP3 is diffusely distributed in the cytosol of uninfected cells, but formed distinct small speck in ZIKV-infected cells (Revised Fig. 2j-l), suggesting that ZIKV facilitates NLRP3 speck formation or activates NLRP3 inflammasome. ASC oligomerization is a direct indicator of inflammasome activation. We have further examined the effects of ZIKV on ASC pyroptosome formation in THP-1-differentiated macrophages. ASC is diffusely distributed in the nucleus and cytosol of uninfected macrophages but formed distinct small speck in infected macrophages (Revised Fig. 2m), and ASC oligomerization is facilitated by ZIKV in infected macrophages (Revised Fig. 2n), suggesting that ZIKV facilitates ASC oligomerization.

Our results also show that in the absence of NS5, the NLRP3 and ASC proteins are co-localized and distributed in the cytosol to form “ring-like” structures [Revised Fig. 6e(a-e)]. In the presence of GFP, the NLRP3 and ASC proteins are still co-localized and distributed in the cytosol to form “ring-like” structures [Revised Fig. 6e(f-j)]. However, in the presence of NS5, the three proteins (NS5, NLRP3, and ASC) are co-localized and distributed in the cytosol to form “sphere-like” structures, in which NS5 (green) locates on inside, NLRP3 (red) locates in middle, and ASC (cyan) locates on outside [Revised Fig. 6e(k-o)].

Please note that overexpression of only inflammasome components (NLRP3 and ASC) induces the formation of “ring-like” structures without stimulus, and however, overexpression of inflammasome components (NLRP3 and ASC) and ZIKV NS5, induces the formation of distinguished “sphere-like” structures. Thus, our results clearly suggest that ZIKV NS5 facilitates the assembly of NLRP3 inflammasome.

Taken together, our results demonstrate that ZIKV infection activates NLRP3 inflammasome by promoting NLRP3 speck formation and ASC oligomerization, and that ZIKV NS5 protein facilitates the assembly of NLRP3 inflammasome.

•Were NS5-NLRP3-ASC co-localization and ring-like structures observed in THP-1 macs stably expressing NS5?

Authors' Response: Thank you for the comment. We agree with you that it would be better to see if NS5-NLRP3-ASC co-localization and ring-like structures are observed in THP-1 macs stably expressing NS5. However, the antibodies for ZIKV E, NLRP3, and ASC proteins used for Immunofluorescence assays are mouse monoclonal antibodies. Therefore, it is technically impossible for us to stain for ZIKV E, NLRP3, and ASC proteins in current conditions and in human cells. However, we demonstrate that ASC speck formation was observed by IFA in ZIKV infected THP-1 macrophages (Revised Fig. 2m). More importantly is that we have demonstrated in many experiments that ZIKV NS5 alone can induce the NLRP3 activation. Our results support that ZIKV infection can induce the activation of NLRP3 inflammasome and the speck formation of NLRP3 and ASC in these cells.

•Was ASC speck formation also observed by IFA in infected THP-1 macrophages?

Authors' Response: Yes, as you suggested, we have determined the effect of ZIKV on ASC speck formation in infected THP-1-differentiated macrophages. The results show that ASC is diffusely distributed in the nucleus and cytosol of uninfected macrophages but formed distinct small speck in infected macrophages (Revised Fig. 2m). Thus, ASC speck formation is also observed in ZIKV infected THP-1 macrophages. In addition, ASC oligomerization is facilitated by ZIKV in infected macrophages (Revised Fig. 2n), suggesting that ZIKV facilitates NLRP3 assembly.

Figure 7: ZIKV infection induces inflammatory responses in mice

- A129 mice lacking IFN a/b receptors were susceptible to ZIKV infection
- Neutrophil/mononuclear cell infiltration in brain
- Corpuscle structures in spleen were large and poorly defined
- Increased vascular permeability in livers and kidneys
- Weight loss began 4 dpi
- All mice died 9-12 dpi
- Authors concluded that ZIKV induced severe inflammatory responses in mice but note the immunocompromised nature of the AG129 mice and one cannot make such

strong conclusions without much more analysis. Are these inflammatory responses dependent on IL-1 β or NLRP3? The authors need to place effort on generating knockout NLRP3 or IL-1 β in A129 mice via breeding with existing NLRP3 and IL-1 β KO lines to properly generate and evaluate inflammatory phenotype in response to ZIKV infection. Otherwise, the observed inflammatory phenotype may simply be driven by cell death or release of other cytokines/DAMPs.

Authors' Responses: Thank you for the comments.

We agree with that it is also very important to demonstrate how ZIKV infection induces severe inflammatory responses in A129 mice with immunocompromised nature. According to your suggestion, we have carried out several additional experiments to determine the effect of ZIKV infection on the induction of inflammatory responses.

Initially, the role of ZIKV NS5 protein in the regulation of IL-1 β activation is evaluated in the cells stably expressed NS5 and treated with a caspase-1 inhibitor (Ac-YVAD-cmk). The results show that IL-1 β secretion (Revised Fig. 4h) as well as IL-1 β and Casp-1 cleavages (Revised Fig. 4i) are induced by NS5, but NS5-induced activations of IL-1 β and Casp-1 are inhibited by Ac-YVAD-cmk, further demonstrating that NS5 facilitates the activation of NLRP3 inflammasome.

In addition, the effects of NLRP3 or IL-1 β on ZIKV induced severe inflammatory responses in A129 mice have been investigated. Many research groups have used A129 mice model to study ZIKV infection and pathogenesis. For examples, ZIKV infection leads to the development of microcephaly in A129 mice (Li et al., *Cell Stem Cell* 2016;19(1):120-6. doi: 10.1016/j.stem.2016.04.017.), and ZIKV causes testis damage and leads to male infertility in A129 mice (Ma et al., *Cell* 2016;167(6):1511-1524.e10. doi: 10.1016/j.cell.2016.11.016.). In this study, we demonstrate that IL-1 β levels are not only significantly higher in the sera of ZIKV infected patients (Revised Fig. 1a), but also increased rapidly in the sera of ZIKV infected A129 mice (Revised Fig. 1c). These results suggest that ZIKV infection plays an stimulatory role in the production of IL-1 β in the blood of human and mice in the absence of type I IFNs.

Moreover, the role of ZIKV infection in the induction of inflammatory responses are evaluated in A129 mice. We demonstrate that ZIKV infection induces inflammatory responses in the brains and spleens of A129 mice (Revised Fig. 7a and b), and induces vascular permeability in the livers and kidneys of A129 mice (Revised Fig. 7d). The body weights of infected mice started to lose weight after 4 days post-infection (Revised Fig. 7g), and infected mice began to die at 9 days post-infection (Revised Fig. 7h).

We agree with you that it would be better to generate knockout NLRP3 or IL-1 β in A129 mice to evaluate inflammatory phenotype in response to ZIKV infection. Considering the time limitation and technique difficulty, we used the caspase-1 inhibitor, Ac-YVAD-cmk, instead of NLRP3 knockout A129 mice in our experiments. Therefore, finally, the effect of ZIKV infection on the induction of inflammatory responses are further determined in A129 mice treated with caspase-1 inhibitor, Ac-YVAD-cmk. We show that in the absence of Ac-YVAD-cmk, IL-1 β levels are induced rapidly by ZIKV infection until 4 days post-infection, but in the presence of Ac-YVAD-cmk, the sera levels of IL-1 β are slightly increased by ZIKV infection (Revised Fig. 8g), suggesting that caspase-1 is required for ZIKV-induced activation of IL-1 β . We also reveal that ZIKV infection induces the production of IL-1b, but such induction is inhibited by Ac-YVAD-cmk, (Revised Fig. 8h), indicating that ZIKV induces the production of IL-1b through regulating the activation of caspase-1. We also reveal that ZIKV infection induces inflammatory responses in the brain, spleen, liver, and kidney of A129 mice, but such inflammatory responses are inhibited by Ac-YVAD-cmk (Revised Fig. 8i), suggesting that ZIKV induces inflammatory responses through regulating the activation of caspase-1. Taken together, we demonstrate that ZIKV induces inflammatory responses through regulating the activation of caspase-1 and the production of IL-1 β .

Finally, the body weights of mice are also reduced by ZIKV infection, but such reduction is attenuated in the presence of Ac-YVAD-cmk (Revised Fig. 8j), and the survival rates of mice are reduced by ZIKV infection, but all infected mice are survived in the presence of Ac-YVAD-cmk (Revised Fig. 8k). Taken together, we

demonstrate that ZIKV infection induces inflammatory responses in mice through the activation of IL-1 β mediated by caspase-1.

Reviewer #2: Thus, overall the study has some major areas of concern and below are suggested places to focus key revisions:

-Repeat experiments showing virus-induced IL-1 β maturation and release in THP-1 macs, PBMCs, or BMDCs using sucrose purified ZIKV that is demonstrated to be mycoplasma-free.

Authors' Responses: Thank you for the comment. The virus was filtered by 0.45 μ m filter for cell infection, and purified by 26% sucrose following harvest for mice infection. We added the experimental procedures to the revised Materials and Method section. The ZIKV stock used in this study has been demonstrated to be mycoplasma-free.

-Show basic virologic analysis of infected PBMCs, BMDCs, or THP-1 macs, including percent cell infectivity and plaque assays showing whether or not infectious viral particles are generated in these cells. Showing viral RNA levels alone is not sufficient.

Authors' Responses: Thank you for the comment. According to your suggestion, we have detected the ZIKV RNA levels in PBMCs, BMDCs, and THP-1 macs by using qRT-PCR (Revised Fig. S1a–d; Revised Fig. 2a and b; Revised Fig. 3c, f, and i), the infectious ZIKV in the supernatant of PBMCs (Revised Fig. S1e) and THP-1 cells (Revised Fig. S1f) by using plaque assays, and the infectious ZIKV in THP-1 cells (Revised Fig. S1g) by using Immunofluorescence assays to detect the ZIKV E protein.

-Show that the NLRP3 inflammasome is necessary for the inflammatory phenotype observed in A129 mice. Include data from wt and NLRP3 ko C57BL/6 mice.

Authors' Responses: Thank you for the comment.

We agree with you that it would be better to generate knockout NLRP3 or IL-1 β in A129 mice to evaluate inflammatory phenotype in response to ZIKV infection. Considering the time limitation and technique difficulty, we used the caspase-1 inhibitor, Ac-YVAD-cmk, instead of NLRP3 knockout A129 mice in our experiments. Therefore, the effect of ZIKV infection on the induction of inflammatory responses are further determined in A129 mice treated with Ac-YVAD-cmk.

Our results show that IL-1 β protein secreted in the sera of infected mice are significantly induced by ZIKV, but such induction is repressed by Ac-YVAD-cmk (Revised Fig. 8g), suggesting that Casp-1 is required for ZIKV-induced secretion of IL-1 β . Similarly, IL-1 β proteins produced in the brain, spleen, liver, and kidney of infected mice were activated by ZIKV, but such activations were inhibited by Ac-YVAD-cmk (Revised Fig. 8h), indicating that Casp-1 is required for ZIKV-induced activation of IL-1 β .

Moreover, the effect of ZIKV infection on the induction of inflammatory responses in mice was evaluated. Inflammatory responses in the brain, spleen, liver, and kidney of infected mice were induced by ZIKV, but such ZIKV-induced inflammatory responses were inhibited by Ac-YVAD-cmk (Revised Fig. 8i), revealing that ZIKV induces inflammatory responses through Casp-1. Finally, the role of ZIKV infection in the growth of mice was evaluated. The body weights of infected mice were reduced during ZIKV infection, and such reductions were attenuated by Ac-YVAD-cmk (Revised Fig. 8j), suggesting that Casp-1 or IL-1 β is important for the infection of ZIKV. The survival rates of mice were reduced by ZIKV infection, and for our surprise, all infected mice were survived in the presence of Ac-YVAD-cmk (Revised Fig. 8k), further confirmed that Casp-1 or IL-1 β is important for the infection of ZIKV. Taken together, we demonstrate that ZIKV infection induces severe inflammatory responses in mice through Casp-1-mediated activation of IL-1 β .

Reviewers' comments:

Reviewer #1 (Remarks to the Author):

My questions have been addressed

Reviewer #2 (Remarks to the Author):

Wang et al present their revised manuscript of their study to assess NLRP3 activation by the Zika virus (zikh) NS5 protein. The authors have worked to address previous comments through inclusion of new data, assessment of any mycoplasma in virus preps, and mostly by rebutting specific concerns raised in the previous review. The new data present are repeats of experiments where now they show totally different results, for example the low IL-1 beta levels induced by zikh are now high IL-1beta levels from what is presented as the same exact experiment design. It is highly concerning that repeating the experiment now gives the outcome that favorable to alleviate the previous comments and the original concern of the reviewer. This occurs in multiple experiments. Also the fold increase of IL-1beta stated by the authors is indeed high (10 to 30 fold) but the level of IL-1beta are not biologically significant in most data sets shown except for where they repeated the experiment in response to reviewer comments and now have much higher level of IL-1beta from the same basic experiment that previously showed little IL-1beta.

The virus stock used is clearly free of mycoplasma, so this concern is no longer relevant and the authors convincingly show that no contamination is present.

However, it is highly concerning to make such strong conclusions on the biology of ZIKV infection as made from a highly compromised mouse model lacking interferon alpha/beta and IFN gamma signaling. The request to validate inflammasome activation in a wt mouse model was directly rebutted by the authors to state that zikh does not infect wt mice. This statement is simply not true - see e.g. the following papers: PMID: 28835502, 27855206, 28231312. These show that zikh does infect wt adult B6 mice, and that like most infections in humans, does not kill the mouse but the infection generates an immune response (T cells, antibodies) that serves to control the spread of the virus. The authors can now use this wt mouse model to assess the induction of IL1-beta production in ZIKV infection during the transient infection.

The conclusions that zikh induces IL-1beta and that this processes is relevant and occurs in human infection to facilitate disease are still not supported by this study. The study nicely shows that zikh can infect ag129 mice and cause disease in the absence of IFN a/b/g signaling.

Point-by-Point Response to the Comments of Reviewer 2

Reviewer 2 Comment: Wang et al present their revised manuscript of their study to assess NLRP3 activation by the Zika virus (zikv) NS5 protein. The authors have worked to address previous comments through inclusion of new data, assessment of any mycoplasma in virus preps, and mostly by rebutting specific concerns raised in the previous review. The new data present are repeats of experiments where now they show totally different results, for example the low IL-1 beta levels induced by zikv are now high IL-1beta levels from what is presented as the same exact experiment design. It is highly concerning that repeating the experiment now gives the outcome that favorable to alleviate the previous comments and the original concern of the reviewer. This occurs in multiple experiments. Also the fold increase of IL-1beta stated by the authors is indeed high (10 to 30 fold) but the level of IL-1beta are not biologically significant in most data sets shown except for where they repeated the experiment in response to reviewer comments and now have much higher level of IL-1beta from the same basic experiment that previously showed little IL-1beta.

Authors' Response: Thank you for your comments. Based on your previous suggestions, we have made lot efforts to address these issues and given full consideration to your concerns. We have clearly demonstrated that the virus stacks were mycoplasma-free and provided enough information of the patients, also repeated very carefully many experiments, including Figure 1d–g; Figure 2d, 2f, 2i, 2j, and 2m; Figure 3h and i; Figure S1a, b, c, f, and g; and Figure S2b, by using several different approaches and methods, and performed many new experiments, including Figure 3f, h, and i, as well as Figure 8, to address your concerns. Some comments that have been supported by other research groups or previous reports have been addressed by citing the relevant publications.

We are sorry for the confusion. Our new results show that ZIKV induces IL-1 β secretion at the concentration of 200 pg/ml in human PBMCs, and the previous results showed that ZIKV induced IL-1 β secretion at the concentration of 120 pg/ml in human PBMCs. 200 pg/ml vs. 120 pg/ml is improved, but is not totally different. The

main reason for the improvement is due to that in the new experiments, the human PBMCs were isolated and cultured for 4 more days and damaged cells were removed to improve the cell viability before infected with the virus.

Our new results also indicate that ZIKV induced IL-1 β secretion at the concentration of 110 pg/ml in mice BMDMs, and the previous results showed that ZIKV induced IL-1 β secretion at the concentration of 15 pg/ml in mice BMDMs. 110 pg/ml vs. 15 pg/ml is significantly improved, but is not totally different. The main reason for such significantly improvement is due to that we purchased new L929 cells and used the supernatants of the new cells for the stimulation of mice BMDMs. We are sorry that we didn't mention in the previous revised manuscript which made you confused. Therefore, although we used the same experiment design, we performed the new experiments more carefully and pay more attention to the details to get better results.

We should clearly that we have only repeated the above two experiments concerning IL-1 β secretion, not multiple experiments, in previous revision.

Reviewer 2 Comment: The virus stock used is clearly free of mycoplasma, so this concern is no longer relevant and the authors convincingly show that no contamination is present. However, it is highly concerning to make such strong conclusions on the biology of ZIKV infection as made form a highly compromised mouse model lacking interferon alpha/beta and IFN gamma signaling. The request to validate inflammasome activation in a wt mouse model was directly rebutted by the authors to state that zikv does not infect wt mice. This statement is simply not true - see e.g. the following papers: PMID: 28835502, 27855206, 28231312. These show that zikv does infect wt adult B6 mice, and that like most infections in humans, does not kill the mouse but the infection generates an immune response (T cells, antibodies) that serves to control the spread of the virus. The authors can now use this wt mouse model to assess the induction of IL1-beta production in ZIKV infection during the transient infection.

Authors' Response: Thank you for your comments.

We appreciate that you agree the virus stocks are mycoplasma-free.

We should clarify that the mice we used in this study is A129 mice lacking IFN- α/β signaling, but not AG129 mice lacking IFN- $\alpha/\beta/\gamma$ signaling.

We agree with your comment that our statement “ZIKV does not infect wt mice” was inaccurate, and have carefully read the three publications that you have provided. Huang et al (PMID: 28835502) reported that ZIKV-infected C57BL/6 WT mice showed mild inhibition of body weight growth, the viral RNA was detected by qRT-PCR in spleen, but not in spinal cord or brain, at day 3 post infection. Moreover, ZIKV was completely undetectable at 7 days post-infection (dpi) and there is no pathogenesis detected. Therefore, ZIKV infects C57BL/6 mice transiently and then might be cleared up by the immune system shortly. Manangeeswaran et al (PMID: 27855206) reported that ZIKV only infects neonatal C57BL/6 WT mice with 1 day old. As indicated by Huang et al (PMID: 28835502) that the immune system of neonatal C57BL/6 WT mice is underdeveloped and is immune compromised. Thus, the neonatal C57BL/6 WT mice are equivalent to the A129 mice in terms of immune compromised. Since the A129 mice model has been widely used and recognized by many research groups to study ZIKV infection and pathogenesis, and thus we previously preferred to use this mice model in our study. Pardy et al (PMID: 28231312) reported that the main reason for them to use C57BL/6 WT mice, but not A129 mice, is the study of T cell response, as type I IFN is required for optimal T cell accumulation. In addition, all the three articles (PMID: 28835502, PMID: 27855206, PMID: 28231312.) agreed with that A129 mice is well-established animal model for the study of the infection and pathogenesis of ZIKV. More importantly, in our study, we indeed demonstrated that ZIKV infection can successfully induce IL-1 β secretion in A129 mice. We strongly believe that A129 mice are good models for our study.

We thank you for your useful comment. Based on your suggestion, we have evaluated the biological effect of ZIKV infection on the induction of inflammatory responses in C57BL/6 WT mice (Please see revised Fig. 7a-c and Supplementary Fig. 7a and b). C57BL/6 WT mice were treated with Ac-YVAD-cmk (caspase-1 inhibitor) and infected with ZIKV as described previously (PMID:28835502). The viral RNA

was detected in the blood of infected mice, and slightly increased in the presence of Ac-YVAD-cmk (Revised Supplementary Fig.7a). Interestingly, IL-1 β protein was induced in the sera of ZIKV-infected C57BL/6 WT mice, but such induction was repressed by Ac-YVAD-cmk (Revised Fig.7a), suggesting that ZIKV activates IL-1 β production and caspase-1 is involved in such activation. The body weights of mock-infected mice were slightly increased during the treatment, whereas ZIKV-infected mice started to lose weight during infection, but the reduction in body weight of ZIKV-infected mice was rescued by Ac-YVAD-cmk (Revised Fig.7b). Moreover, ZIKV infection induced splenomegaly and significant increased splenic index in the mice, but such inductions were repressed by Ac-YVAD-cmk (Revised Fig.7c). The viral RNA replication was confirmed in the spleen of infected mice, and slightly increased in the presence of YVAD-cmk (Revised Supplementary Fig.7b). Thus, our new results suggest that ZIKV infection may induce inflammatory response in C57BL/6 WT mice.

Reviewer's Comment: The conclusions that zikv induces IL-1beta and that this processes is relevant and occurs in human infection to facilitate disease are still not supported by this study. The study nicely shows that zikv can infect ag129 mice and cause disease in the absence of IFN a/b/g signaling.

Authors' Response: Again, thank you for the comment.

In fact, many research groups have used A129 mice model to study ZIKV infection and pathogenesis. For examples, ZIKV infection leads to the development of microcephaly in A129 mice (Li et al., Cell Stem Cell 2016; 19(1):120-6. doi: 10.1016/j.stem.2016.04.017. PMID:27179424.), ZIKV causes testis damage and leads to male infertility in A129 mice (Ma et al., Cell 2016; 167(6):1511-1524.e10. doi: 10.1016/j.cell.2016.11.016. PMID:27884405.), and more recently A129 mice model has been used to study ZIKV infection (Yuan et al. A single mutation in the prM protein of Zika virus contributes to fetal microcephaly. Science 2017, pii: eaam7120. doi: 10.1126/science.aam7120. [Epub ahead of print]). A129 mice were also used to evaluate ZIKV vaccine (Shan et al., Nat Med 2017; 23(6):763-767. doi:

10.1038/nm.4322. Epub 2017 Apr 10. PMID: 28394328.).

In this study, we not only demonstrate that ZIKV infects A129 mice and causes diseases in the absence of IFN- α/β signaling in Figures 7 and 8, but also reveal that IL-1 β levels are significantly higher in the sera of ZIKV infected human patients and that ZIKV infection induces IL-1 β secretion in human PBMCs and differentiated human THP-1 cells in Figures 1, 2 and 3. Moreover, we also demonstrate that ZIKV NS5 protein interacts with NLRP3 and facilitates the NLRP3 inflammasome assembly and activation in Figures 4, 5 and 6.

Thank you for your suggestion. Now, we also demonstrate that ZIKV infection induces inflammatory response in C57BL/6 WT mice in the revised manuscript. We show that in C57BL/6 WT mice, ZIKV infection activates IL-1 β production and induces inflammatory responses, and Casp-1 is involved in such activations.

Taken together, we believe that ZIKV infection activates IL-1 β secretion, induces inflammatory responses, and may promote disease development.

REVIEWERS' COMMENTS:

Reviewer #2 (Remarks to the Author):

The authors have addressed by concerns in part by stating that the larger differences in IL-1beta production shown in the previous version of the manuscript (15 pg/ml to 110 pg/ml) are not really different. This type of reasoning is not assuring to the reviewer, and it is not known why the authors now get high levels of IL-1beta when addressing my comments whereas before my comments they actually got low levels of IL-1beta in their experiments.

All concerns have been addressed in this way.

Point-by-point Response to the Issue Raised by Reviewer #2

Reviewer #2 (Remarks to the Author):

The authors have addressed by concerns in part by stating that the larger differences in IL-1beta production shown in the previous version of the manuscript (15 pg/ml to 110 pg/ml) are not really different. This type of reasoning is not assuring to the reviewer, and it is not known why the authors now get high levels of IL-1beta when addressing my comments whereas before my comments they actually got low levels of IL-1beta in their experiments.

All concerns have been addressed in this way.

Authors Response:

We are sorry for the confusion.

Our new results indicate that ZIKV induced IL-1 β secretion at the concentration of 110 pg/ml in mice BMDMs, and the previous results showed that ZIKV induced IL-1 β secretion at the concentration of 15 pg/ml in mice BMDMs. 110 pg/ml vs. 15 pg/ml is significantly improved. The main reason for such significantly improvement is due to that we purchased new L929 cells and used the supernatants of the new cells for the stimulation of mice BMDMs. Therefore, although we used the same experiment design, we preformed the new experiments more carefully and play more attention to the details to get better results.

We appreciate your suggestion, which has made the result more clear.